

A trait-based modelling approach to planktonic foraminifera ecology

Maria Grigoratou[1], Fanny M. Monteiro[1], Daniela N. Schmidt[2], Jamie D. Wilson[1], Ben A. Ward[1, 3] and Andy Ridgwell[1, 4].

[1]School of Geographical Sciences, University of Bristol, Bristol BS8 1SS, UK
[2]School of Earth Sciences, University of Bristol, Bristol BS8 1RL, UK
[3] Ocean and Earth Science, University of Southampton, National Oceanography Centre, Southampton, European Way, Southampton SO14 3ZH, UK
[4]Department of Earth Sciences, University of California, Riverside CA, Geology Building, 900 University Ave, Riverside, CA 92521, USA

[*]*Correspondence to*: Maria Grigoratou (maria.grigoratou@bristol.ac.uk)

Abstract:

Despite the important role of planktonic foraminifera in regulating the ocean carbonate production and their unrivalled value in reconstructing paleoenvironments, our knowledge on their ecology is limited. A variety of observational techniques such as plankton tows, sediment traps and experiments, have contributed to our understanding of foraminifera ecology. But, fundamental questions around costs and benefits of calcification, and the effect of nutrients, temperature and ecosystem structure on these organisms remain unanswered. To tackle these questions, we take a novel mechanistic approach to study planktonic foraminifera ecology based on trait theory. We develop a 0-D trait-based model to account for the biomass of prolocular (20 μm) and adult (160 μm) stages of non-spinose foraminifera species and investigate their potential interactions with phytoplankton and other zooplankton under different temperature and nutrient regimes. Building on the costs and benefits of calcification, we model two ecosystem structures to explore the effect of resource competition and temperature on planktonic foraminifera biomass. By constraining the model results with ocean biomass estimations of planktonic foraminifera, we estimate that the energetic cost of calcification could be about 25-50% and 20-35% for prolocular and adult stages respectively. Our result suggest that the shell provides protection among predation (e.g pathogens protection) and that the invariably low standing biomass of planktonic foraminifera plays a key role in their survival from predation, along with their shell protection. Temperature appears to be an important factor in regulating foraminifera biomass in the early developmental stage, whereas resource competition is a key in controlling adults' biomass and feeding strategy.



## 1. Introduction

Planktonic foraminifera as a group comprise fifty holoplanktonic heterotrophic protozoans (Kucera, 2007). They are the most widely-used zooplankton group to reconstruct past marine environments, with proxies devised that are based on their abundance, assemblage composition, and/or physio-geochemical characteristic of their shell (e.g Schmidt et al., 2003; Schiebel and Hemleben, 2005). They are also the most important calcifying zooplankton group, supplying between 23-55% of the total marine planktonic carbonate production (Schiebel, 2002), and hence are a key contributor to the composition of marine sediments (Schiebel and Hemleben, 2005).

In contrast to their high abundances in sediments, they tend to grow at very low abundance in the ocean and never dominate the zooplankton community, representing less than 5% of total microprotozooplankton abundance (Beers and Stewart, 1971). Based on plankton tow observations, abundances range from 1 ind. $m^{-3}$ in blue waters, 20-50 ind. $m^{-3}$ in oligo- and mesotrophic waters (Schiebel and Hemleben, 2005) to >1000 ind. $m^{-3}$ in polar regions (Volkmann, 2000). Their global biomass in the water column has been estimated to be between 0.002 and 0.0009 PgC and their contribution to global plankton biomass to be ~ 0.04% (Buitenhuis et al., 2014).

Despite their importance in palaeo- and modern biochemical oceanography, our knowledge of planktonic foraminifera's physiology, development and ecology is limited to observations of a few species. Planktonic foraminifera are difficult to grown in culture and it has been impossible to culture a next generation (Schiebel and Hemleben, 2017). Consequently, information regarding the intraspecies and interspecies competition as well as a mechanistic understanding of their ecology through their whole life cycle is missing. The development and application of numerical ecological models can help fill in this knowledge gap, of which a particularly promising approach which we focus on here, involves consideration of physiological traits and their associated trade-offs.

Trait-based approaches can be useful for improving our knowledge of planktonic foraminifera ecology as they canaddress fundamental questions around the cost of growth across developmental stages, their position in the global food webs and calcification. Trait-based approaches provide a mechanistic understanding of individuals, populations or ecosystems as they describe these systems from first principles by highlighting key traits (e.g. feeding, competition, predation, reproduction) and associated trade-offs (e.g Litchman and Klausmeier, 2008; Litchman et al., 2013; Kiørboe, 2008; Barton et al., 2016; Hébert et al., 2016). For example, body size is considered as a master trait for plankton, impacting many physiological and ecological aspects such as metabolic rates (e.g. growth), diet, abundance, biomass and reproduction (e.g Litchman et al., 2013).

A number of traits and trade-offs have been identified for planktonic foraminifera, summarised in Figure 1. The shell size of planktonic foraminifera can be regarded as a 'master' trait and can be used as an indicator for growth optimal environmental conditions. Planktonic foraminifera development is divided into five stages, defined based on shell size and wall structure: prolocular, juvenile, neanic, adult and terminal (gametogenesis) (Brummer et al., 1986, 1987). Their shell diameter ranges from about 10 μm for the prolocular life stage to more than 1250 μm for the adult under optimal conditions (Schmidt et al., 2004a). Planktonic foraminifera are considered to reach the adult stage and subsequently be sexually mature when their shell size reaches 100 μm (Brummer et al., 1986). Shell size increases from low to high latitudes (Schmidt et al 2003, 2004b) and is related to reproductive success (gametogenesis), as bigger individuals release more gametes (e.g. Caron and Bé, 1984; Hemleben et al., 1987). Temperature and food availability are suggested to be the main environmental factors which regulate their size (e.g. Anderson et al., 1979; Spero et al., 1991; Caron et al., 1983;



Schmidt et al., 2004a), but a mechanistic understanding of the response of shell size to temperature and food is missing.

Calcification is another important trait of planktonic foraminifera, relative to shell size, but the costs and benefits of their shell and the nature of the associated trade-off are not well understood.
Paleo records indicate size, thickness, and morphology changes of planktonic foraminifera shell responds to changing climates (Keller and Pardo, 2004). Given the impact of climate change on the ocean and its ecosystems, determining the cost and benefit of producing a shell is fundamental to quantifying the influence of climate change on planktonic foraminifera ecology, distribution, and carbonate production in the past, present and future.

The feeding strategies of planktonic foraminifera are also an important trait as they are crucial for survival and influence plankton community ecology. Planktonic foraminifera are inactive organisms and passive feeders. They do not detect their prey but encounter them while drifting, using a rhizopodial network which extends from their body (e.g Anderson and Bé, 1976). As planktonic foraminifera are typically collected for culturing at sizes >60 μm and subsequently grown as individuals, information
regarding the feeding behaviour of the early (prolocular and juvenile) life stages, the cost and benefits of being inactive passive feeder, and interactions with other plankton are missing. It has been suggested that at the prolocular stage all species are herbivorous (Schiebel and Hemleben, 2017) and subsequently widen their food sources. Field and lab observations suggest that spinose species use their spines, which start growing during the juvenile stage, to capture and control active zooplankton
prey, that are often larger than them (e.g., Anderson, 1983; Spindler et al., 1984). Species of this group tend to be either omnivorous or carnivorous (Schiebel and Hemleben, 2017). Most spinose species develop a symbiotic relationship with photosynthesizing algae (Schiebel and Hemleben, 2017) and are dominant in oligotrophic areas. It has been speculated that the higher abundance is due to their carnivorous feeding as these areas are characterized by relative low phytoplankton concentration and
relative high abundance of copepods (Schiebel et al., 2004; Moriarty and O'Brien 2013). Non-spinose species are considered to be omnivorous/herbivorous (Anderson et al., 1979; Hemleben and Auras, 1984), with the ability to catch and feed on small zooplankton or dead organic matter. These species have low abundance in oligotrophic areas and their maximum in high-productivity regions (Schiebel and Hemleben, 2017).

Trait-based models can supplement the physiological and ecological understanding of foraminifera gained in the field and cultures (Fig. 1) and be used as a complimentary method to go through culture limitations and improve our understanding of planktonic foraminifera's ecology. Trait-based models have been successfully applied to phytoplankton (e.g Follows et al., 2007; Litchman and Klausmeier, 2008; Monteiro et al., 2016) with little development and application on zooplankton (e.g.
(Banas, 2011; Maps et al., 2011; Ward et al., 2012; 2014; Banas et al., 2016). However, until now, a few species models have been developed to study the ecology of modern planktonic foraminifera species: Žarić et al. (2006) (from now on Žarić06), PLAFOM (Fraile et al., 2008; Fraile et al., 2009) and FORAMCLIM (Lombard et al., 2011; Roy et al., 2015). Žarić06 develop an empirical model which relates the global fluxes of eighteen species of planktonic foraminifera to environmental conditions based on
observations of eighteen species. PLAFOM model field observations to predict the influence of temperature (Fraile et al., 2008) and food availability (Frail et al., 2009) on the global biogeography of five species. FORAMCLIM represents eight species of planktonic foraminifera and studies the influence of temperature, food availability, light and climate change on growth rates and global distribution. These models provide important insights regarding the interaction between planktonic foraminifera
and their habitat. Their main limitation is that are based on either empirical data (Žarić et al., 2006;



Fraile 2008; 2009) or growth data from laboratory (Lombard et al., 2011; Roy et al., 2015) and their application is thus species-specific and limited to specific environmental ranges (Roy et al., 2015).

Here, we describe the first trait-based generic model of planktonic foraminifera using body size, calcification and feeding behaviour as key traits to investigate the mechanisms behind planktonic
foraminifera ecology. We focus on modelling non-symbiotic non-spinose species as a starting point because these species are predominantly herbivorous throughout their whole life, and do not develop spines and algal symbionts, all of which increase complexity and are not sufficiently constrained by basic physiological data. Our trait-based planktonic foraminifera model was derived from the size-structured plankton models of Ward et al. (2012; 2014) which use cell and body size as the eco-
physiological trait to study the phyto-zooplankton food web. With this model, we investigate the energetic costs and benefits of calcification, their feeding behaviour and resource competition with other zooplankters, as well as the environmental controls on two different developmental stages. Model results assess and quantify the biotic and abiotic factors influencing their physiology and ecology, and the interactions of planktonic foraminifera, with phytoplankton and other zooplankton,
as well as their environment.

## 2. Methods
### 2.1. Model environment
Our model represents a chemostat experiment in a zero-dimensional (0D) setting. It accounts for one source of nutrients (here defined as $NO_3^-$) and fifty-one generic phytoplankton and zooplankton size classes from pico- to mesoplankton (Schiebel 1978).

The nutrient availability ($NO_3^-$) depends on the input nutrient concentration ($N_o$) interpreted as either a nutrient-rich vertical source of nutrient (typical of high-productivity regions) or a less-rich
horizontally advective nutrient source (typical of oligotrophic gyres), dilution rate $\kappa$ and phytoplankton uptake (Eq. (1)).

$$\frac{dN}{dt} = \kappa * (N_o - N) - \sum_{j_{prey}=1}^{J} \left[ B_{N,j} \right] P_{growth,j} \tag{1}$$

We investigated a range of $N_o$ values (0-5 mmol N m$^{-3}$) to account for a range of different nutrient regimes, from oligotrophic to eutrophic (Ward et al., 2014). We assumed that the terms of plankton mortality (plankton loss due to viral/bacterial infection or natural death) and zooplankton sloppy feeding (prey which is lost from the predator during feeding (Latwon, 1970)) are exported out of the chemostat. There is no nutrient recycling in the model.

Plankton populations are modelled in terms of nitrogen biomass $\left[ B_{N,j} \right]$ with the rate of change of biomass described by:

$$\frac{dB}{dt} = \left[ B_{N,j} \right] P_{growth,j} + \left[ B_{N,j} \right] \sum_{j_{prey}=1}^{J} \lambda_{i_{b,j}} \; G_{N \, jprey} - \sum_{j_{pred}=1}^{J} \left[ B_{N,jpred} \right] G_{N \, jpred,j} - \left[ B_{N,j} \right] m_j \tag{2}$$

where $P_{growth,j}$ represents the phytoplankton growth, $\sum_{j_{prey}=1}^{J} \lambda_{i_{b,j}} \; G_{N \, jprey}$ the zooplankton growth, $\sum_{j_{pred}=1}^{J} \left[ B_{N,jpred} \right] G_{N \, jpred,j}$ the plankton losses due to zooplankton grazing and $m_j$ plankton background mortality. The model parameters and symbols are defined in Tables 1 and 2 and a more detailed description of the model and plankton growth is available in Appendix A.



### 2.2. Complexity of the ecosystem structure

We modelled two simplified ecosystems: a simple food chain and a more complex food web (Fig. 2). In the simple food chain model, zooplankton are herbivorous size-specialist predators feeding on one prey size group. In order to examine the grazing pressure of a specialist predator on planktonic foraminifera, we made an exception by defining one zooplankton group to be omnivorous, capable of consuming only planktonic foraminifera and one phytoplankton group with the same size as planktonic foraminifera. Resource competition occurs mostly at the phytoplankton level. In zooplankton, the only competition is between individual planktonic foraminifera and with zooplankton of the same size group (Fig. 2a). This simple representation of the marine ecosystem allows us to better understand the model behaviour and the top-down and bottom-up controls on foraminifera while testing the grazing pressure of a specialist predator on planktonic foraminifera.

In the food web model, resource competition occurs at both phytoplankton and zooplankton levels. Zooplankton predators are size-generalist omnivorous predators able to consume more than one prey (Fig. 2b). This more complex version helps us to better understand how the herbivorous non-spinose planktonic foraminifera can compete with other omnivorous zooplankters and handle multi predation pressure. The food web model has a more realistic representation of the plankton community but the dynamic interactions within the groups are more challenging to disentangle (Banas 2011; Ward et al., 2014). With the two versions of the model we are able to examine how the resource competition within plankton community as well as predation, influences different life stages of planktonic foraminifera.

The switch from the food chain to food web version is implemented through predators' grazing kernel, which dictates the relative palatability of potential prey (fig. 3, Eq. (3)). In this parameterization, the prey palatability ($\varphi_{jpred, j\,prey}$) expresses the likelihood of a predator to eat a prey (eq. 3) and it depends on the optimum predator:prey length ratio ($\theta_{opt}$), the log size ratio of each predator with each prey ($\theta_{jpred, jprey}$), and the standard deviation ($\sigma$) which shows the width of size prey preference and defines how specialist or generalist the predator can be (Fig. 3).

$$\varphi_{jpred, j\,prey} = \exp\left[-\left(ln(\frac{\theta_{jpred, jprey}}{\theta_{opt}})\right)^2 \left(2\sigma_{jpred}^2\right)^{-1}\right] \tag{3}$$

We assumed a 10:1 predator:prey length ratio as the optimum size for zooplankton to feed upon, as is often observed for zooplankton (Kiørboe, 2008). Prey with a size ratio equal to this optimum therefore have the highest prey palatability of this particular predator. For the food chain model, predators can only consume one prey group that was exactly ten times smaller than themselves ($\sigma = 0.0001$). In the food web model, we allow zooplankton to be more generalist predators and feed on prey of size around this optimum ratio but with a smaller palatability to acknowledge that zooplankton can feed on prey of a wider size range (Kiørboe, 2008) ($\sigma = 0.5$). When considering generalist planktonic foraminifera (foob web model), we tested a range of different grazing kernels ($\sigma = 0.5 - 1.0$). This is because the model results showed that being more generalist than other zooplankton groups is a condition for planktonic foraminifera to survive.





### 2.3. Adding planktonic foraminifera in the model

*Planktonic foraminifera biomass*

We compared our modelled biomass to observations from Schiebel and Movellan (2012) and Buitenhuis et al. (2014), converting from, PgC m$^{-3}$ to mmolN m$^{-3}$, using the carbon molecular weight (12 gC/mol) and an assumed Redfield C:N stoichiometry of 6.625. We assumed that there is no correlation between the species and the size fractions of Schiebel and Movellan's (2012) samples and

we estimated that the relative biomass of the non-spinose planktonic foraminifera 150- 200 μm size fraction to micro- and mesozooplankton biomass ranges from 0.02% (5x10$^3$ mmolNm$^{-3}$) to 0.03% (1x10$^4$ mmolNm$^{-3}$). Due to the lack of data, we presumed that the prolocular biomass is similar to the adult biomass. We extended the biomass range to be from 0.01% to 0.09% in order to include a global biomass representation for early stages and small adults based on Schiebel and Movellan (2012)

suggestion that biomass of early stages can be up to three times higher than adults with size <125 μm. Model simulations for which planktonic foraminifera relative biomass is within the observed range of 0.07% to 0.09% are referred here as 'low biomass'.

*Calcification*

With the model we tested basic hypotheses to investigate the trade-offs of shell size and calcification and the effect of resource competition on planktonic foraminifera biomass for two life stages, prolocular (20 μm) and the adult (160 μm). Each life stage was modelled independently. As the costs and benefits of foraminifera's calcification are not experimentally known, we added a calcifying zooplankton type in the model with an associated trade-off for calcification, following the Monteiro

et al. (2016) representation of a calcifying phytoplankton type (coccolithophore). To model non-spinose planktonic foraminifera, we use the same parameterization and equations as for zooplankton, hypothesizing that the main cost for shell development is energy loss, and the main benefit of calcification is protection (Monteiro et al., 2016). Preliminary experiments showed that the background mortality ($m$) had to be decreased to keep planktonic foraminifera at low biomass,

suggesting that the shell may act as a protection against other factors than predation (e.g. pathogens, parasites).

Studies have shown that zooplankton metabolic rate and biomass varies with temperature (Ikeda, 1985), but the reasons behind the correlation between habitat and mortality rate are still very complicated to quantify (Aksnes and Ohman, 1996). To estimate the cost and benefit of calcification

we ran a sensitivity analysis by decreasing planktonic foraminifera maximum growth ($G_{max}$) and background mortality ($m$) from 0% to 95% (in 5% steps), representing calcification's energy loss and benefit. As there are no data to compare our calcification results with, we selected the model simulations with a variation of 0% to 30% of the reduction in maximum growth and background mortality for all tested environments as most likely (herein denotedas 'plausible' simulations).

In the end, to quantify the benefit of predation protection, we chose a number of 'plausible' simulations to examine different predation pressures on planktonic foraminifera by decreasing the grazing term $\left( G_{N_{jpred,jprey}} \, , \; eq(S3) \right)$ by 100% (no grazing pressure on planktonic foraminifera), 75%, 50%, 25% and 0% (no protection from grazing pressure) of its initial value.




### 2.4. Model set up and numerical simulations

We explored the potential ecological controls on planktonic foraminifera ecology by means of a series of ensembles of model experiments (Table 3). Each individual ensemble was designed to explore a wide range of potential parameter value combinations of growth, predation and background mortality rates and hence different trade-off assumptions and growth conditions. The ensembles were repeated for different potential assumed ecological structures and life stages (prolocular and adult) of planktonic foraminifera. We then apply a series of 'plausibility' filters on the model results to derive
a series of sub-sets of experiments that we analyze in detail and discuss the implications of.

        We ran experiments for nine different environmental combinations; with three input nutrient concentrations ($N_o$= 1, 2.5 and 5 mmol N m$^{-3}$) to represent oligo-, meso- and eutrophic environments respectively and three water temperatures (10$^o$C, 20$^o$C, 30$^o$C). Every experiment was run for 10,000 days (~27 years) until steady state (biomass ± 0.01 mmolNm$^{-3}$). For the food web version, the majority
of the experiments reach an oscillatory steady state close to an equilibrium which is still present after running the model for more than 270 years (results not shown). This oscillatory behaviour is a common feature in ecosystem models (e.g. Baird et al., 2010) especially of planktonic communities (e.g. Petrovskii and Malchow 2001a; Petrovskii et al., 2001; Banas et al., 2011). The initial concentration of all plankton groups was set to 0.0001 mmol N m$^{-3}$.
We present the absolute and relative biomass of planktonic foraminifera from all tested scenarios of calcification costs and benefits in supplementary materials (SM) based on the last 1000 days of the simulations. From 921 (500 for the food chain and 421 for the food web) tested simulations 9.5% (88 simulations) were within the 'low biomass' criteria. From the 'low biomass' simulations, 75% (64 simulations) cover the conditions of the 'plausible' criterion. Due to the low number of 'plausible'
simulations (<4) per environment (SM, figures 4-7), we were not able to perform statistical analysis and instead we provided ranges of values for costs and benefits of calcification in non-spinose planktonic foraminifera for each life stage. We ran 100 simulations for both stages and model versions to examine different predation on planktonic foraminifera.

## 3. Results

### 3.1. General plankton distribution at different environments

        Both versions of the model showed an increasing diversity and biomass from oligo- to eutrophic environments and from cold to warmer environments (Fig. B1) capturing the main patterns of marine
plankton dynamics (e.g Irigoien et al., 2004; Müren et al., 2005; O'Connor et al., 2009). In the food chain version, biomass of phytoplankton and zooplankton increased continuously with the number of coexisting size groups (Fig. B2a). In contrast, the food web version had a patchy distribution of biomass with fewer coexisting groups, equivalent to "winners" of resource competition, and an overall lower biomass than the food chain model (Fig. B2b) similar to previous studies (e.g. Armstrong et al., 1994;
Banas et al., 2011).

        Pico-, nanophytoplankton and nano- microzooplankton dominated the plankton biomass at 10$^o$C in both versions (Fig. B1b) as they outcompete the larger cell sizes through resource competition. As the concentration of the incoming nutrients ($N_o$) was increased from oligo- to eutrophic the growth rate and coexistence of phytoplankton groups also increased, leading to a higher grazing pressure of
zooplankton, biomass and zooplankton co-existence. In the food chain model, microphytoplankton survived in the eutrophic environment at low temperatures (10$^o$C) and all the nutrient environments



at 20℃ and 30℃ model. In the food web, microphytoplankton were present in meso- and eutrophic environments at 20℃ and 30℃. Mesozooplankton were sustained in meso- and eutrophic environments at 20℃ for the food chain model, in eutrophic environments at 20℃ for the food web
model, and in the all environments at 30℃ at both versions of the model (Fig. B1b). Since our model captured the general trends of plankton community through different environments, we used it interrogate the importance of individual traits and trade-offs.

### 3.2. Planktonic foraminifera ecology
**3.2.1. Cost of calcification**

We estimated the potential energetic cost of calcification in non-spinose planktonic foraminifera. In the food chain model, of the 500 simulations, 10.6% (54 simulations) were within the 'low biomass' criterion and 8% (39 simulations) 'plausible'. The 'plausible' simulations resulted in a decrease of
foraminifera growth rate by 10 to 30% for the prolocular stage and 10 to 20% for the adult stage (Figs. 4, 5). For the adult stage, we found no 'plausible' simulations for the environment of mesotrophic at 20℃ due to high decrease of the background mortality (>60%) compare with their low decrease (10%) of growth rate.

Of the 421 food web simulations, 8% (34 simulations) were 'low biomass' and 6% (25 simulations)
'plausible'. The biomass of the prolocular stage increased with temperature and nutrients. The model could not produce any 'low biomass' simulation of early life stages of foraminifera at 30℃ as values were significantly too high (1-7.3% of the total zooplankton biomass, Fig. 6). In all environments at 10℃ and for oligotrophic environments at 20℃ the 'plausible' simulations showed a 10-35% decrease of growth rate. To maintain the prolocular biomass within the defined low biomass range in meso-
and eutrophic environments at 20℃, the calcification cost was equal to a 50% reduction of the growth rate (Fig. 6). The model did not generate results for adults in oligotrophic waters at 10℃ as only small zooplankton groups (<63 μm) could survive for that ecosystem and no 'plausible' simulations for the eutrophic ecosystem at 20℃ and 30℃, as planktonic foraminifera relative biomass was higher than the defined range (Fig. 7). For all the other ecosystems the 'plausible' simulations resulted in a cost of
calcification for the adult stage ranged from 10-45% (Fig. 7).

**3.2.2. Potential benefits of calcification in planktonic foraminifera**

Both versions of the model showed that in order to maintain planktonic foraminifera within the
defined biomass range, the background mortality rate of both prolocular and adult stages had to be reduced by 10-40% (Figs. 4- 7). Our results suggest that planktonic foraminifera use their shell not only for predation protection but for other reasons e.g against pathogens like bacteria or viruses.

Regarding the use of the shell as protection from predation, both model versions showed different results. This is due to different feeding behaviour of zooplankton (specialist vs generalist) as in both
models, predation depends on the feeding behaviour of the predator, prey size and biomass.

In the food chain model, the foraminifera biomass could be maintained inside the observed range when grazing pressure was reduced by 25% for the prolocular and 50% for the adult stage compared to full predation (Fig. B3). Therefore, both low biomass and possession of hard parts are important mechanisms against specialist predators.

Shell protection against predation had no effect on the relative low biomass of foraminifera in the food web model as their biomass remained the same with or without predation at both life stages





(Fig. B3). The food web version suggests that for a generalist predator low biomass is a more efficient protective mechanism than the shell. We found that with a combination of higher than observed biomass of planktonic foraminifera and a predation pressure lower than 50%, planktonic foraminifera
became a dominant group with up to 22% of the total zooplankton biomass suggesting that the shell has a protective function (results not shown).

### 3.2.3. Temperature and feeding control amongst different life stages of planktonic foraminifera

We focus on the results of the food web as it considers resource competition between planktonic foraminifera and the rest of zooplankton and simulates the plankton food web better than the food chain. Our model suggested that being herbivorous is a successful strategy for the prolocular stage as their optimum size prey group (≈2-3µm, as determined by the 10:1 predator:prey size ratio) was present in high abundance in all environments (Fig. 8). Resource competition is therefore not a
determinant factor for the prolocular stage. The model results suggest that temperature had a stronger control on this stage, resulting in higher biomass (1-7%) at 30°C (Supplementary Material).
        Adult foraminifera in the model achieved realistic relative biomass only when they became more generalist feeders by increasing their prey palatability by 20% ($\sigma$= 0.6) for meso- and eutrophic conditions and by 80% ($\sigma$= 0.8) to 100% ($\sigma$ =1.0) in oligotrophic environments (relatively to $\sigma$= 0.5 for
other zooplankton) (Fig. 9). Without this change, adult herbivorous foraminifera in the model were out-competed by omnivorous predators. To understand if feeding behaviour or the lower growth rate and mortality associated with calcification led them to become more generalists, we switched the feeding behaviour in the model from herbivorous to omnivorous. The results showed that omnivorous planktonic foraminifera did not need to be more generalist than the other zooplankters (results not
shown). Resource limitation had therefore an important role in controlling for the non-spinose planktonic foraminifera adult stages.

### 4. Discussion

        We developed the first size-based 0D model of two life stages (one prolocular, 20 µm and one
adult, 160 µm) of planktonic non-spinose foraminifera, to investigate the cost and benefits of calcification and feeding behaviours under different environmental conditions (temperature and nutrient). It is important to note that the present model, like other size structured models, cannot capture the complexity of the plankton community (Banas, 2011) but represents general patterns and encapsulates basic physiological relationships. The model shows that diversity increases from oligo-
to eutrophic environments, and from cold to warmer environments. The model therefore captures the increase in complexity in planktic ecosystems toward the topics and eutrophic systems (Irigoien et al., 2004).
        In the ocean, phytoplankton biomass and productivity are controlled by nutrient availability, light, temperature and grazing pressure (Irigoien et al., 2004). In oligotrophic areas, nutrient limitation leads
to the dominance of small size phytoplankton cells as there is not enough energy to sustain larger cells (Menden – Duer and Kiørboe, 2016). As nutrient availability increases, phytoplankton size diversifies. Zooplankton shows similar pattern; oligotrophic environments are dominated by small heterotrophs, while the size of the species increases in eutrophic environments (Razouls et al., 2018). Our model captured this general pattern, but it struggled to sustain a high biomass of the largest size groups of
microphytoplankton and metazooplankton especially in non-eutrophic environments. We suggest that the oversimplification of physiological and behavioural traits especially for zooplankton leads to





this limitation, as species are represented as spheres with fixed half-saturation ($K_N$) and assimilation efficiency ($\lambda$) (more details in supplementary methods). Changing the shape of the body from a sphere towards an eclipse for representing metazoans, combined with variable half-saturation, may
circumvent this problem. Other aspects which are not represented such as feeding motility, an important trait for organisms' survival (e.g feeding, predation protection) with strong influence on metabolic rates (e.g Ikeda, 1985) could also improve model results.

In the present study we tried to quantify the cost and benefit associated with calcification in planktonic foraminifera. Our model suggests a cost of calcification in non-spinose planktonic
foraminifera of 10-50% for the early life stages and adults. This cost is similar to estimates for coccolithophores (~30%; Monteiro et al., 2016) and for shell production of marine benthic molluscs (22-50%; Palmer, 1992). While biocalcification evolved in the Precambrian and across many clades, metabolic costs may be comparable as pathways and constraints are similar for a range of organisms (Knoll 2003).

Our model results suggest that planktonic foraminifera calcify for a combination of reasons (e.g. protection from pathogen, parasites and grazers), as suggested by other studies on calcifying phytoplankton (Hamm et al., 2003; Hamm and Smetacek 2007; Monteiro et al., 2016). In the light of limited physiological data, we interpret the function of the foraminifera shell to be pathogen and predation protection at both developmental stages. Observations show that bacteria can attack the
cytoplasm of unhealthy or dead planktonic foraminifera (Schiebel and Hemleben, 2017). More field and laboratory studies are needed in order to compare our findings and gain a deeper knowledge on the interaction between planktonic foraminifera and pathogens.

The question about selective predators in planktic foraminifers is still not well understood. While benthic foraminifers are selectively preyed upon by scaphopods (Murray, 1991), evidence for
predation on foraminifera is limited for the planktonic ones. For the neanic and juvenile stages of planktonic foraminifera it is difficult to detect their remains in faecal pallets due to their small size, thin walls and low biomass and as so there are no published data available (Schiebel and Hemleben, 2017). Adults shell and spines have been detected mostly in faecal pallets of metazooplankton groups (like salps, copepods, pteropods, and euphausiids) and nekton shrimps (Be et al., 1977; Bradbury et
al., 1970; Berger, 1971b). Our results highlight that low biomass is a main mechanism for protection against predation in foraminifers.

The food web model results showed that reducing grazing pressure could be a potential benefit of calcification for planktonic foraminifera if they were to become more abundant. The earliest planktonic foraminifera are thin shelled and very small (Gradstein et al., 2017), while modern species
have more complex morphologies with larger and thicker shells (Schmidt et al., 2004). While the planktonic ecosystem became more complex over the last 150 Ma, we speculate that their low abundance and thick shells may have prevented the evolution of a specific predator in contrast to other dominant phytoplankton groups with shell like diatoms (Hamm et al., 2003; Hamm and Smetacek 2007). Based on the results of our model and our current knowledge on foraminiferal
physiology, we propose that the combination of low abundance and a carbonate shell protect planktonic foraminifera against predation. As planktonic foraminifera are non-mobile, it difficult for predators to sense them (Kiørboe, 2008; Van Someren Gréve et al., 2017). Their thick shell can then act as an armour when a grazer reaches them to counter-balance their non-motility. Planktonic foraminifera are thus high-energy-demand prey: they are hard to find and digest, corroborating earlier
suggestions that foraminifera do not have specific predators (Hemleben et al., 1989). We suggest that





planktonic foraminifera non-mobility is an important behavioural trait to be further tested in order to improve our understanding of -grazing protection.

Temperature and food appear to be the main controlling factors of planktonic foraminifera ecology and distribution in the ocean (e.g. Ortiz et al.,1995; Bé and Tolderlund 1971) corroborated by modelling studies (Žarić et al., 2006; Frail et al., 2008, 2009; Lombard et al., 2009; Roy et al., 2015). Studies have shown that sea surface temperature (SST) is one of the most important environmental factors of planktonic foraminifera's diversity (Rutherford et al., 1999) and size (Schmidt et al., 2006;2004a). Field observations (e.g. Bé and Tolderlund 1971), geochemical analysis (Elderfield and Ganssen, 2000) and culture experiments (Caron et al., 1987a, b) show that adult species have a specific optimum temperature range which control their size development and abundance (Schmidt et al., 2004a; Žarić et al., 2005; Lombard et al., 2009). In the present study, we use our trait-based model to study planktonic foraminifera as a group of species to investigate the general patterns of the influence of temperature and resource on planktonic foraminifera biomass on both juvenile and adult stages.

We find that temperature is the main limiting factor for the prolocular life stage, since there is no food limitation. Our model provides insights on the importance of resource availability and competition during development, resulting in a switch to generalist herbivorous and omnivorous at adult stages. Food availability impacts planktonic foraminifera ecology (e.g. Ortiz et al., 1995; Schmidt et al. 2004a) and culture experiments highlight that the amount and type of food (phytoplankton and or zooplankton) have a strong influence on growth rate (e.g. Spindler et al., 1984; Anderson et al., 1979), shell size (Bé et al.,1981) and gametogenesis (Caron et al., 1981; Caron and Bé, 1984; Hemleben et al., 1987). The model results support the hypothesis that during early stages planktonic foraminifera have a herbivorous diet. It also indicates that food availability is a key controlling factor of the biomass of non-spinose adult stages and defines their type of feeding strategy for different nutrient concentration environments.

We propose that non-spinose adult planktonic foraminifera are a very successful herbivorous predator, capable to prey on different phytoplankton size groups or that they can be omnivorous and use other food sources like bacteria, detritus and zooplankton. Observations suggest an opportunistic feeding behaviour for non-spinose species. Diatoms are usually considered to be their primary prey (e.g. Spindler et al., 1984; Hemleben et al., 1985) though some can also consume dinoflagellates (e.g. Anderson et al., 1979), and cryophytes which are either slowly digested or used as symbionts (Hemleben et., 1989). Animal tissues have been found in several non-spinose species (Anderson et al., 1979; Hemleben and Spindler, 1983a). *Globorotalia menardii*, the most abundant and biggest non-spinose species, is suggested to actively control microzooplankton (ciliates) prey (e.g Hemleben et al., 1977). Culture experiments suggest cannibalism between non-spinose but never between spinose species (Hemleben et al, 1989). These observations support our results that non-spinose adult species can feed on different types and size of phytoplankton or switch to omnivory and use other resources when phytoplankton concentrations are rare.

Our model provides important information on how resource competition among planktonic foraminifera and other zooplankters influences the feeding behavior of different life stages and their distribution. Moreover, the inability of our model to sustain non-spinose foraminifera in warm oligotroph regions agrees with observations as planktonic foraminifera are dominated by symbiont bearing species in these regions (Bé and Tolderlund, 1971). Our model results can provide new perspectives regarding the development of symbiosis as an additional energy source in planktonic foraminifera and hence adding symbiosis in the model can be a next important step for improving our understanding of planktonic foraminifera ecology.



## 5. Conclusions

This study takes a first step towards including planktonic foraminifera ecology as part of the plankton community in a trait-based framework and estimates the energetic cost of calcification and the associated benefits. We find that the energetic cost of calcification for both prolocular and adult stages varies between 10-30% in the food chain model and 10-50% in the food web model. Considering the function of the shell, we consider that both low biomass and the carbonate shell are key elements for protection of planktonic foraminifera from predation. A reduction in mortality by 15-40% suggest that the shell may be more important for pathogens and parasites than against grazing pressure.

Similar to coccolithophores (Monteiro et al., 2016), the costs and benefits of calcification in planktonic foraminifera vary with the environments. In the model, temperature is the dominant factor for the prolocular stage, whereas both temperature and resources are important for the adult. Consequently, the adults are more impacted by resource competition driven by less available food in the optimal size of their prey and resulting in the need to feed on a wider range of prey size. This finding is particularly relevant in oligotrophic environments where food is scarce. We therefore suggest that the adults are generalist herbivorous or omnivorous or use other resources in oligotrophic environments such as symbiosis.

To develop the model further, data on energy allocated to growth, calcification and mobility is needed to better understand the physiology and ecology of this important paleoclimate proxy carrier and producer of marine carbonates. Other traits and trade-offs such as feeding mechanism (rhizopodial network, spines), mobility, symbiosis with algae need to be tested in the future and supported by culture experiments.

## Code access

The code can be found online at the supplement materials, https://doi.org/10.5281/zenodo.1487877.

## Author contributions

MG, FMM and DNS designed the study. MG, JDW and BAW developed the model. MG prepared the manuscript. All authors contributed to writing and editing the final version of the manuscript.

## Competing interests

The authors declare that they have no conflict of interest.

## Acknowledgements

This work was supported by the European Research Council "PALEOGENiE" project (ERC-2013-CoG617313). This work was also supported by NERC (grant number NE/J019062/1) to FMM. DNS would like to acknowledge support via a Wolfson Merit Award from the Royal Society.

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



**Table 1**. Model parameters (Ward et al., 2014 and references with in).

| Parameter | Symbol | Value or formula | Units |
|---|---|---|---|
| Deep N concentration | $N_0$ | Variable (0-5) | $mmol\ N\ m^{-3}$ |
| Chemostat mixing rate | $\kappa$ | 0.01 | $day^{-1}$ |
| Light limitation | $\varphi$ | 0.1 | - |
| Optimal predator: prey length ratio | $\theta_{opt}$ | 10.0 | - |
| Standard deviation of $log_{10}(\theta)$ | $\sigma_{graz}$ | 0.001[*], 0.5[a], 0.6[b], 0.8[c], 1[d] | - |
| Total prey half- saturation | $K_N^{prey}$ | 0.1501 | $mmol\ N\ m^{-3}$ |
| Assimilation efficiency | $\lambda$ | 0.7 | - |
| Prey refuge parameter | $\Lambda$ | -1 | $mmol\ N\ m^3$ |
| Phytoplankton mortality | $m_P$ | 0.02 | $day^{-1}$ |
| Zooplankton mortality (food web) | $m_z$ | 0.02 | $day^{-1}$ |
| Zooplankton mortality (food chain) | $m_z$ | $0.05V^{-0.16}$ | $day^{-1}$ |
| Maximum growth rate at 20°C | $\mu_{max}$ | $\dfrac{P_C^{max}\ V_N^{max}\ \Delta Q}{V_N^{max}\ Q_N^{max} + P_C^{max} Q_N^{min}\ \Delta Q}$ | $day^{-1}$ |
| Half- saturation for growth | $K_N$ | $\dfrac{P_C^{max}\ K_{NO_3}\ Q_N^{min}\ \Delta Q}{V_{NO_3}^{max}\ Q_N^{max} + P_C^{max}\ \Delta Q}$ | $mmol\ N\ m^{-3}$ |

[*]: value for the simple food chain, [a]: zooplankton and prolocular stage of planktonic foraminifera, [b]: adult stage of planktonic foraminifera for meso- and eutrophic ecosystems, [c,d]: adult stage of planktonic foraminifera for oligotrophic ecosystem of 20°C and 30°C respectively.






**Table 2.** Size- dependent parameters (adapted from Ward et al., 2012, see references within). Coefficients a and b are used in the power-law function that assigns parameters as a function of plankton cell volume $p = aV^b$.

| Parameter | Symbol | a | b | Units |
|---|---|---|---|---|
| Maximum photosynthetic rate | $P_{C,prochlorococcus}^{max}$ | 1.0 | -0.15 | $day^{-1}$ |
|  | $P_{C,synechococcus}^{max}$ | 1.4 | -0.15 | $day^{-1}$ |
|  | $P_{C,other}^{max}$ | 2.1 | -0.15 | $day^{-1}$ |
|  | $P_{C,diatoms}^{max}$ | 3.8 | -0.15 | $day^{-1}$ |
| Maximum nitrogen uptake rate | $V_{NO_3}^{max}$ | 0.51 | -.027 | $day^{-1}$ |
| Half -saturation for uptake | $K_{NO_3}$ | 0.17 | 0.27 | $mmol\ N\ m^{-3}$ |
| Phytoplankton minimum N quota | $Q_N^{mim}$ | 0.07 | -0.17 | $mmol\ N\ (mmol\ C)^{-1}$ |
| Phytoplankton minimum N quota | $Q_N^{max}$ | 0.25 | -0.13 | $mmol\ N\ (mmol\ C)^{-1}$ |
| Maximum prey capture rate | $G_{max}$ | 21.9 | -0.16 | $day^{-1}$ |






**Table 3**: Summary of studied traits and environmental conditions for the non-spinose planktonic
foraminifera.

| Plankton interactions | | |
|---|---|---|
| Model version | Structure | Plankton size groups |
| food chain | One prey per predator<br>Zooplankton: passive, herbivorous<br>Planktonic foraminifera: passive, herbivorous | 25 phytoplankton |
| food web | Multi prey per predator<br>Zooplankton: passive, omnivorous<br>Planktonic foraminifera: passive, herbivorous | 25 zooplankton<br><br>1 planktonic foraminifera |

| Environmental Conditions | | | | |
|---|---|---|---|---|
| Model version | Temperature (°C) | 10 | 20 | 30 |
| food chain & food web | Nutrient region | O<br>O<br>O | M<br>M<br>M | E<br>E<br>E |

| Study traits | | | |
|---|---|---|---|
| Model version | Shell size | Calcification | |
| | | Trait | Trade-off |
| food chain & food web | Prolocular (20 µm) | Energy loss | Protection from predation & pathogens/parasites (mortality term) |
| food chain & food web | Adult (160 µm) | Energy loss | Protection from predation & pathogens/parasites (mortality term) |

| Main outcomes | | | | | |
|---|---|---|---|---|---|
| Model version | Shell size | Calcification | | | temperature & resource control (results based on the food web) |
| | | Energy loss (%) | Protection | | |
| | | | predation | % morality reduce | |
| food chain | Prolocular (20 µm) | 10-30 | Shell & low biomass [*] | 10-40 | Temperature |
| food web | | 10-50 | low biomass [**] | | |
| food chain | Adult (160 µm) | 10-20 | Shell & low biomass [*] | 10-40 | Resource |
| food web | | 10-45 | low biomass [**] | | |

[*]The model showed that both shell and low biomass are important for protection from predation.
[**]The results showed that low biomass is more important for protection from predation.






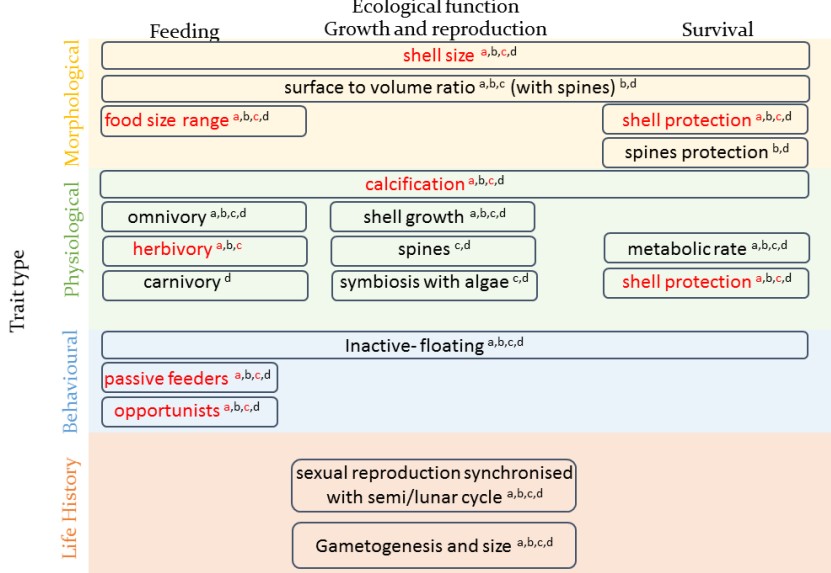

**Figure 1**. Schematic presentation of planktonic foraminifera traits and tradeoffs. The examined traits of the present study are shown in red






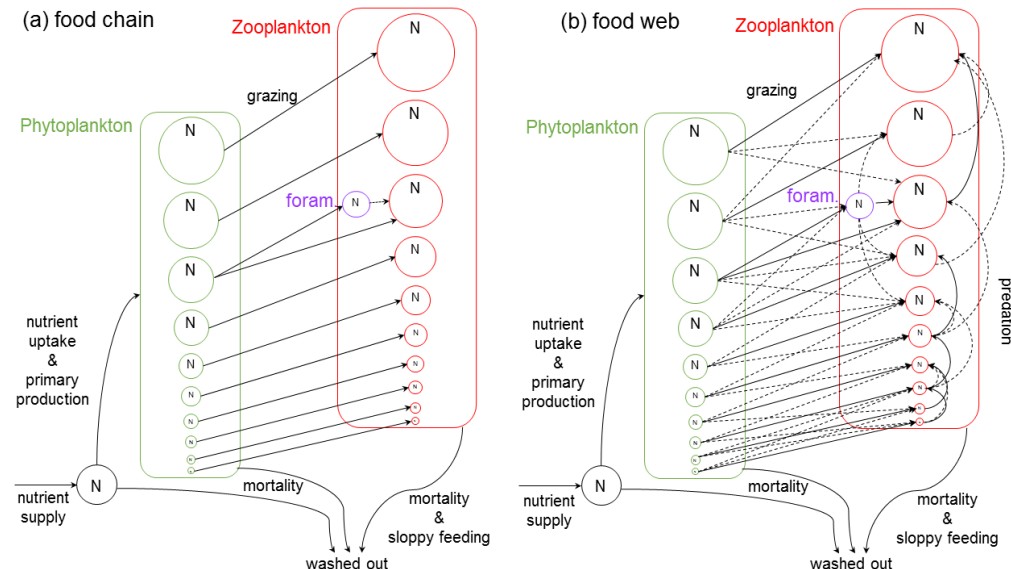

**Figure 2.**: Schematic description of the two model versions of the size-trait-based model of planktonic
foraminifera: **(a)** food chain; and **(b)** food web (adopted with permission from Ward et al., 2012).




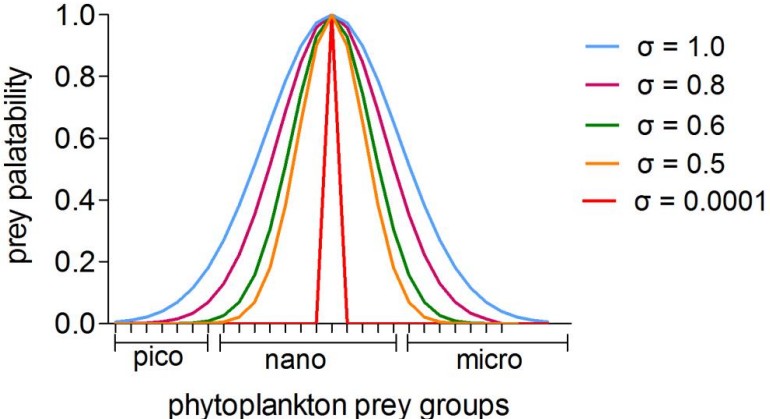

**Figure 3.** Illustration of prey palatability trend with phytoplankton prey groups for one example of herbivorous
predator (160 μm size). Size specialist predator (present in the food chain version) is characterised by σ = 0.0001.
Size generalist predator (present in the food web version) is characterised by σ ≥ 0.5.




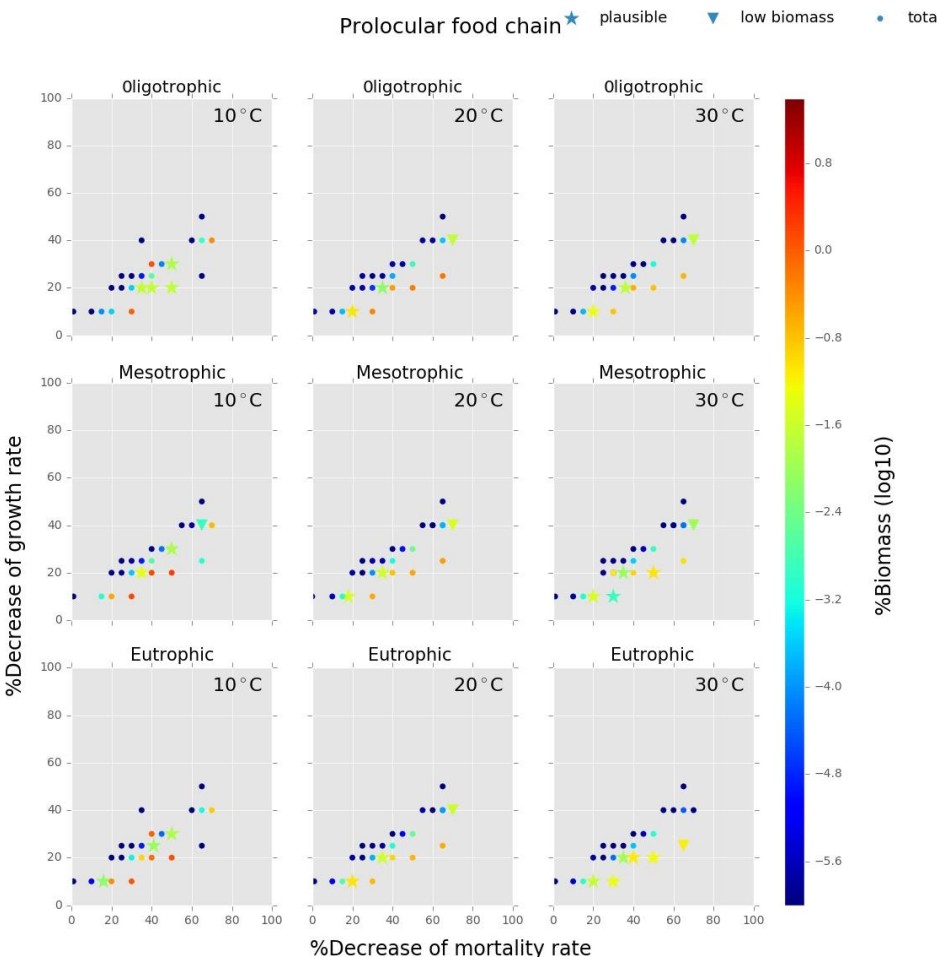

**Figure 4.** Results from the food chain model for the calcification cost (reduction of growth) and benefit (reduction of mortality rate) for the prolocular life stage of planktonic foraminifera. Legend shows 'total' for total tested simulations, 'low biomass' for simulations for which their biomass is within the defined range, and 'plausible' for the simulations we consider to be as most likely. More details for 'low biomass' and 'plausible' simulations in the Methods, section 2.3: adding planktonic foraminifera into the model.




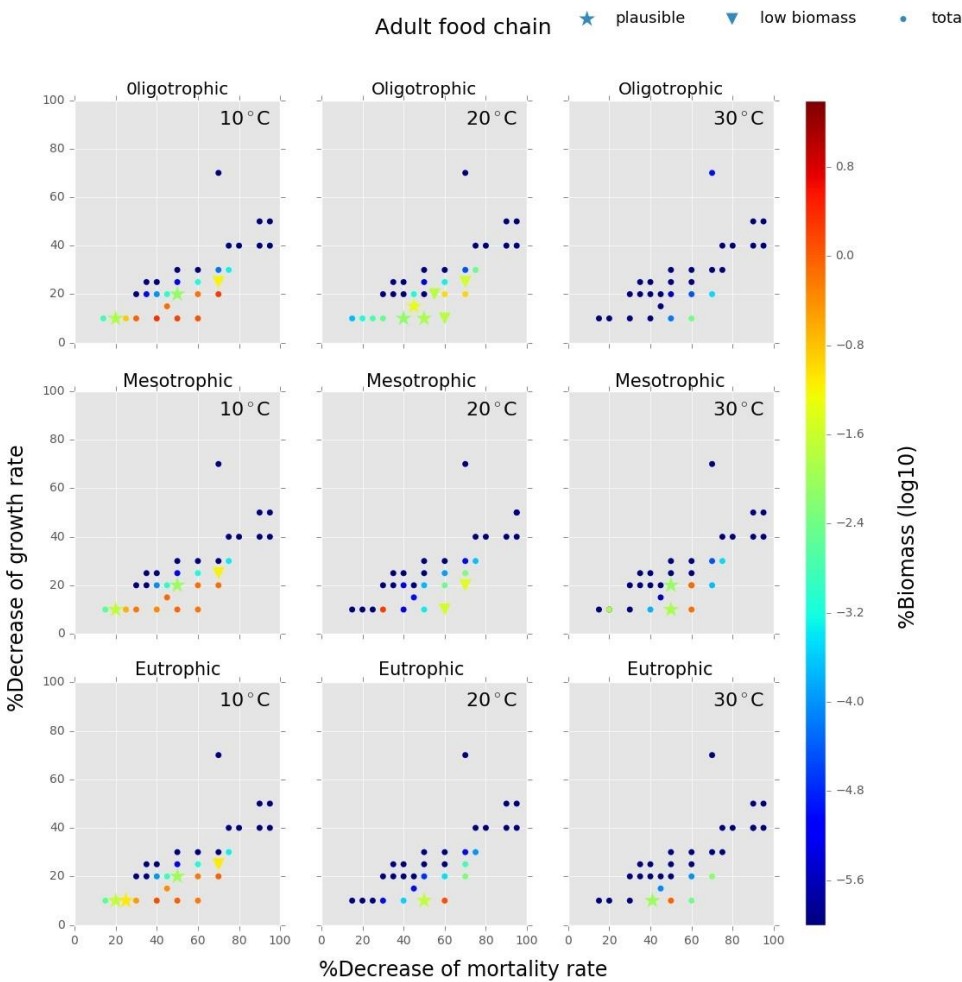

**Figure 5.** Results from the food chain model for the calcification cost (reduction of growth) and benefit (reduction of mortality rate) for the adult life stage of planktonic foraminifera. Legend shows 'total' for total tested simulations, 'low biomass' for simulations for which their biomass is within the defined range, and 'plausible' for the simulations we consider to be as most likely. More details for 'low biomass' and 'plausible' simulations in the Methods, section 2.3: adding planktonic foraminifera into the model.




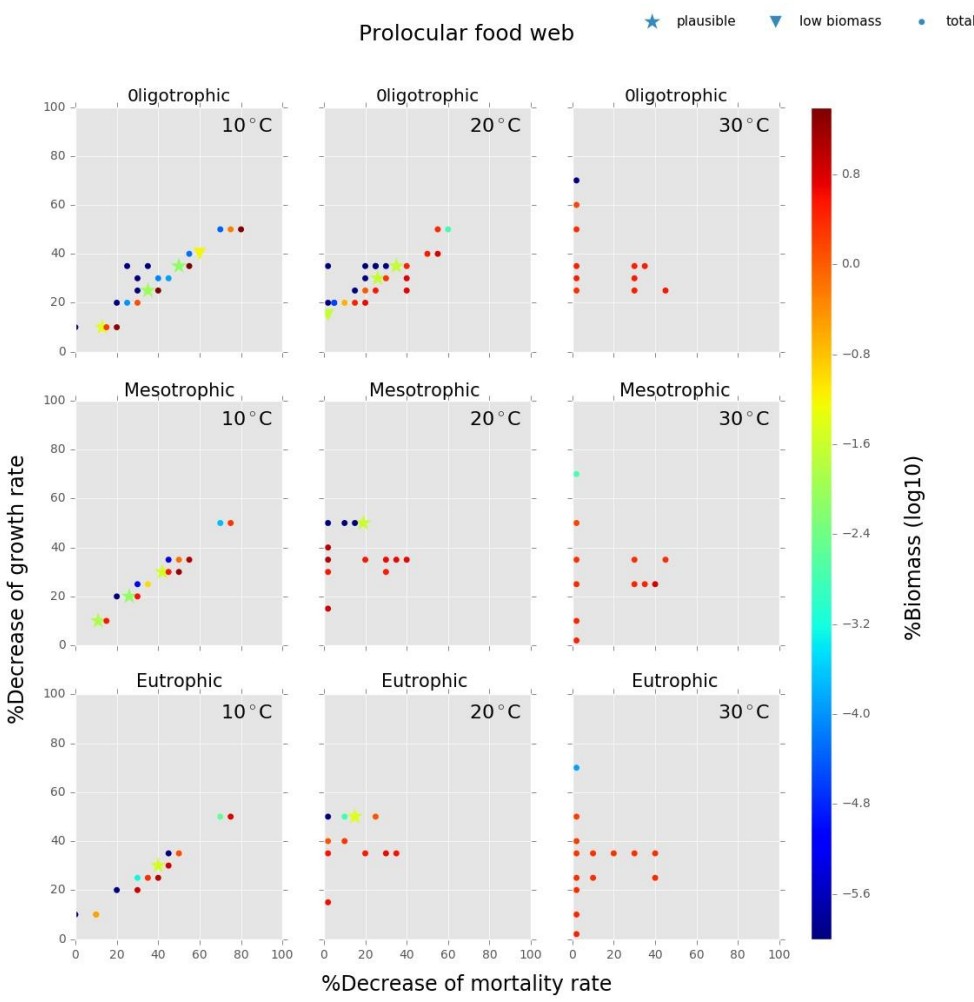


**Figure 6.** Results from the food web model for the calcification cost (reduction of growth) and benefit (reduction of mortality rate) for the prolocular life stage of planktonic foraminifera. Legend shows 'total' for total tested simulations, 'low biomass' for simulations for which their biomass is within the defined range, and 'plausible' for the simulations we consider to be as most likely. More details for 'low biomass' and 'plausible' simulations in the Methods, section 2.3: adding planktonic foraminifera into the model.




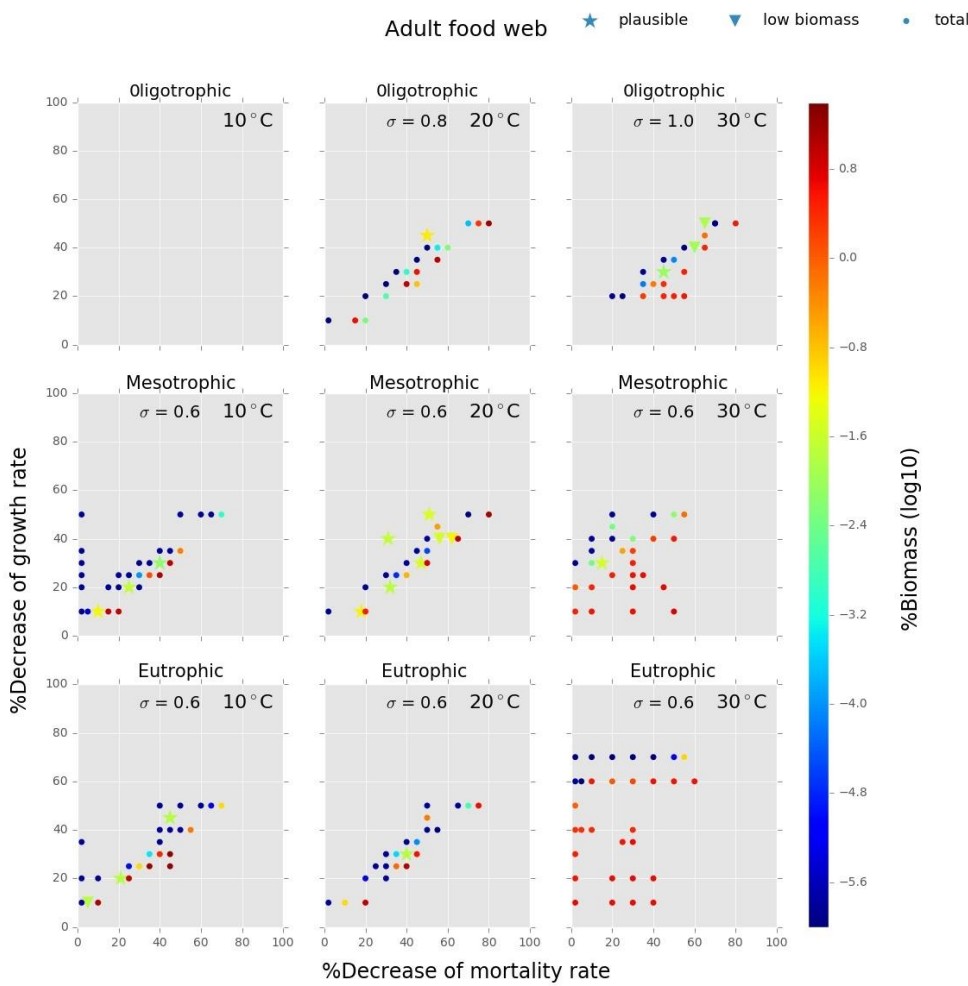

**Figure 7.** Results from the food web model for the calcification cost (reduction of growth) and benefit (reduction of mortality rate) for the adult life stage of planktonic foraminifera. Legend shows 'total' for total tested simulations, 'low biomass' for simulations for which their biomass is within the defined range, and 'plausible' for the simulations we consider to be as most likely. More details for 'low biomass' and 'plausible' simulations in the Methods, section 2.3: adding planktonic foraminifera into the model.





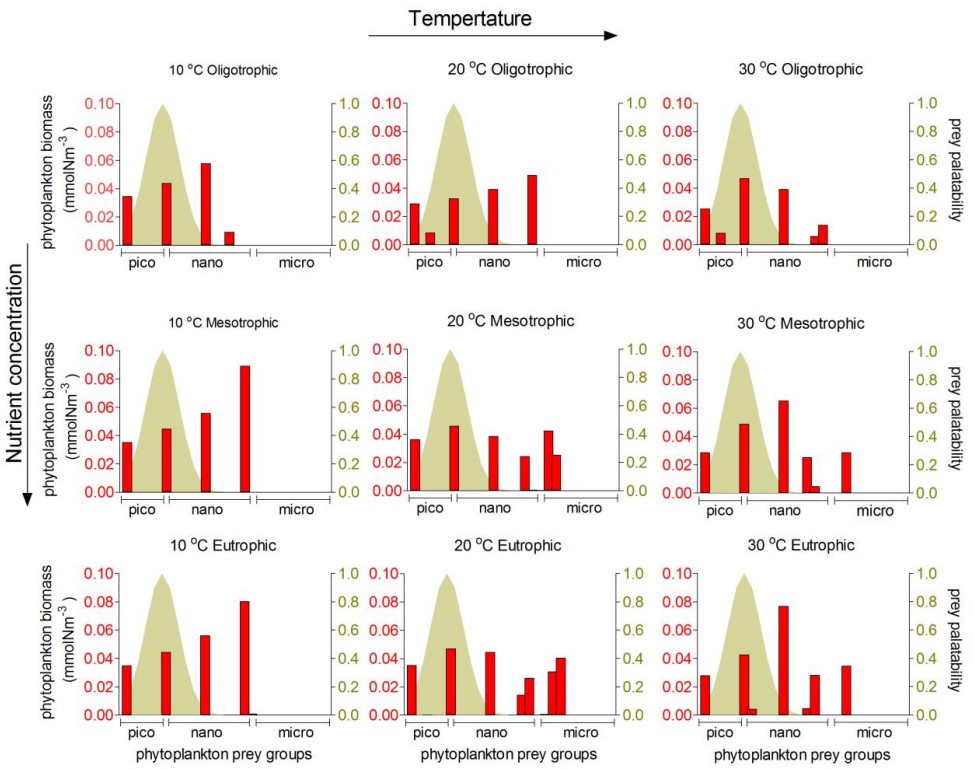

**Figure 8**. Model results of resource competition for the prolocular stage (20 μm) of planktonic foraminifera in the food web version. Left axis (red columns): biomass (mmolN m$^{-3}$) of phytoplankton size groups. Right axis (colored shadow): prey palatability of planktonic foraminifera using a σ = 0.5.




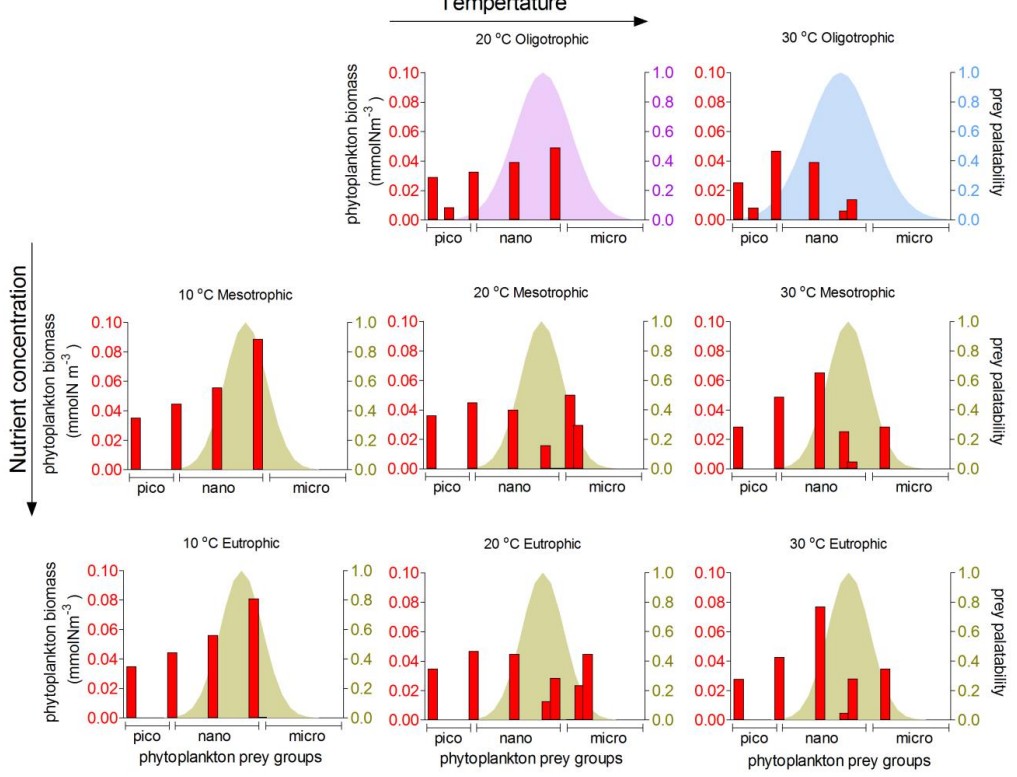

**Figure 9**. Model results of resource competition for the prolocular stage (20 μm) of planktonic foraminifera in
the food web version. Left axis (red columns): biomass (mmolN m$^{-3}$) of phytoplankton size groups. Right axis
(colored shadow): prey palatability of planktonic foraminifera. For oligotrophic enviroments, σ = 0.8 and 1 for
20°C and 30°Crespectively.  For all meso- and eutrophic ecosystems σ = 0.6. No zooplankton larger than 100 μm
and adult stage of planktonic foraminifera survived in the oligotrophic ecosystem at 10°C for the model set up.







**Appendix A**

**Model description**

*Plankton size groups*

        We selected plankton cell sizess in the model so that the volume of each plankton doubles from
one class to another (similar to Ward et al., 2014). We set up the model to have six pico- (0.6-2.0 µm),
ten nano- (2.6- 20 µm) and nine micro- groups (25-160 µm) for the phytoplankton; and six nano- (6-
20 µm), ten micro- (26- 200 µm) and nine (250- 1600 µm) meso- groups for the zooplankton. The
diagnostic equation for plankton biomass (phytoplankton and zooplankton) is given in $eq(1)$ and
shows the generic dependence of biomass with nutrient uptake, zooplankton grazing and mortality.
The symbols are explained in Tables 1 and 2.

*Environmental variables*

        The model accounts for two environmental variables influencing plankton growth: light and
temperature. Light limitation is represented as a fixed parameter set to 0.1 (equivalent to 90% of light
limitation; Ward et al., 2014). The influence of temperature on plankton metabolic rates ($\gamma_T$ ) is
represented by an Arrhenius-like equation (Eq. A1) with  ($T_{ref}$) the reference temperature at which
$\gamma_T = 1$ is 293.15 K (20°C).

$$\gamma_T = e^{R(T-T_{ref})} \tag{A1}$$

We tested three ambient water temperatures ($T$) : 10, 20 and 30°C characteristic of subpolar,
subtropical and tropical regions respectively. Phytoplankton maximum growth rate ($\mu_{max}$) has been
normalised to 20°C (Table 1); and the temperature limitation is represented by $\gamma_T$. Temperature has
a proportionate impact on both phytoplankton and zooplankton growth (eqs S2, S3).

*Phytoplankton*

        Phytoplankton growth ($P_{growth,j}$) is size-dependent and described via Monod equation assuming
there is a balance between the nutrient uptake and growth of phytoplankton (Monod, 1950) (Eq. A2).

$P_{growth,j} = \frac{\mu_{max}*N}{N+K_N} * \varphi * \gamma_T$     (A2)

        Phytoplankton half- saturation ($K_N$) and maximum specific growth rate ($\mu_{max}$) are also cell-size
dependent (Table 1). The maximum uptake rate ($\mu_{max}$) is a function of the maximum photosynthetic
rate ($P^{max}$), the cell volume ($V_N^{max}$) and the phytoplankton quota (Tables 1 and 2) (Ward et al., 2014).
The maximum photosynthetic rate ($P^{max}$) and the phytoplankton quota (Tables 1 and 2) (Ward et al.,
2014). The maximum photosynthetic rate ($P^{max}$) for each size class of phytoplankton is represented
by observations (Irwin et al., 2006) of Prochlorococcus for the two first pico- groups (0.6 and 0.8 µm)
and of *Synechococcus* for the rest four pico- groups, other eukaryotes for nano- and diatoms for
microphytoplankton (Table 2).






*Zooplankton*

We used the zooplankton grazing term as has been described in Ward et al., 2012 with two different feeding behaviours of zooplankton. Zooplankton grazing ($G_{N_{jpred,jprey}}$) is represented using
the Holling type II function (Eq. A3). We choose Holling type II as it describes a decelerating increase of predator ingestion rate to prey concentration consistent with what is observed for most zooplankton (Kiørboe et al., 2018). Although most of zooplankton have different feeding behaviours in different life stages, Holling type II better illustrates predator-prey relationships of many ambush zooplankton groups in the lab over a long-term period (Kiørboe et al., 2018).


$$G_{N_{jpred,jprey}} = G_{max} * \gamma_T * \frac{\varphi_{jpred,jprey}*B_N}{F_{N,j_{pred}}+K_N^{prey}} * \text{Prey refuge}_{N,j_{prey}} * \Phi_{P\vee Z} \qquad (A3)$$

Zooplankton growth:
Zooplankton grazing depends on maximum grazing rate and prey palatability $\varphi_{jpred,j\ prey}$ (Eq.3).
The maximum grazing rate ($G_{max}$) is size dependent (Table 2). The prey palatability ($\varphi_{jpred,j\ prey}$) express the likelihood of a predator to eat the prey. It depends on the log size ratio of each predator with each prey ($\theta_{jpred,jprey}$) with the optimum predator:prey length ratio ($\theta_{opt}$).
The total prey biomass available to each predator ($F_{N,j_{pred}}$) is calculated as a sum of prey biomass weighted by their prey palatability (Eq. A4).


$$F_{N,j_{pred}} = \sum_{jprey=1}^{J} \varphi_{j_{pred,jprey}} \left[ B_{N_{jprey}} \right] \qquad (A4)$$

We set the zooplankton half-saturation constant ($K_N^{prey}$) to 0.1051 mmol N m⁻³. This value is a conversion of Ward et al. (2012) value (1 mmol C m⁻³) from carbon to nitrogen based on Redfield ratio.
While observations show evidence of a variable half-saturation constant for zooplankton (e.g. Hansel et al., 1997), there is not enough information to tease apart its value for the different species, so we assumed a constant K among our zooplankton groups.
We also included a prey-refuge term in the model using the Mayzaud and Poulet's function (1978) (Eq. A5). The prey-refuge term describes how the grazing rate of the predator changes with prey
density and never satiates (Gentleman and Neuheimer, 2008). At high prey density the grazing rate is similar to Holling type I where it becomes linearly related to the prey availability ($\Phi_{P\ or\ Z}$) (eq (S7)). When the prey density is low, the decay constant parameter ($\Lambda$) decreases the grazing pressure such as the grazing rate is similar to Holling type III without any saturation (Gentleman and Neuheimer, 2008). Planktonic foraminifera are the only group with no prey refuge term to account the cost of their
inability to escape predation (Kiorboe et al., 2008).

$$\text{Prey refuge}_{N\ jprey} = \left( 1 - e^{-\Lambda F_{N,j_{pred}} * \Phi_{PorZ}} \right) \qquad (A5)$$

We allowed zooplankton in our model to switch feeding behaviour from filter herbivorous to
ambush carnivorous ($\Phi_{P\ or\ Z}$) as a function of the prey's biomass and size ((Gentleman et al., 2003; Ward et al., 2012) (Eq. A6, A7).





$$\Phi_P = \frac{\sum_{j_{phyto}=1}^{J} \varphi_{jpred,jphyto}\left[B_{N_{jphyto}}\right]^2}{\sum_{j_{prey}=1}^{J} \varphi_{jpred,jprey}\left[B_{N_{jprey}}\right]^2} \qquad (A6)$$


$$\Phi_Z = \frac{\sum_{j_{zoo}=1}^{J} \varphi_{jpred,jzoo}\left[B_{N_{jzoo}}\right]^2}{\sum_{j_{prey}=1}^{J} \varphi_{jpred,jprey}\left[B_{N_{jprey}}\right]^2} \qquad (A7)$$

Finally, we assumed a size-dependent mortality term for zooplankton in the food chain model because there is no zooplankton predation on zooplankton (Table 1) (Ward et al., 2014). As in the food
web model there is predation of zooplankton, we assumed a linear mortality term equal to phytoplankton (Table 1) (Ward et al., 2012).

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



**Appendix B**

945          In the Appendix B, we present model results of the plankton biomass (Figure B1), the
coexistence of plankton size groups in different nutrient environments (Figure B2) and the examples
of planktonic foraminifera's shell protection against different predation pressures in the food chain
and food web (Figure B3).





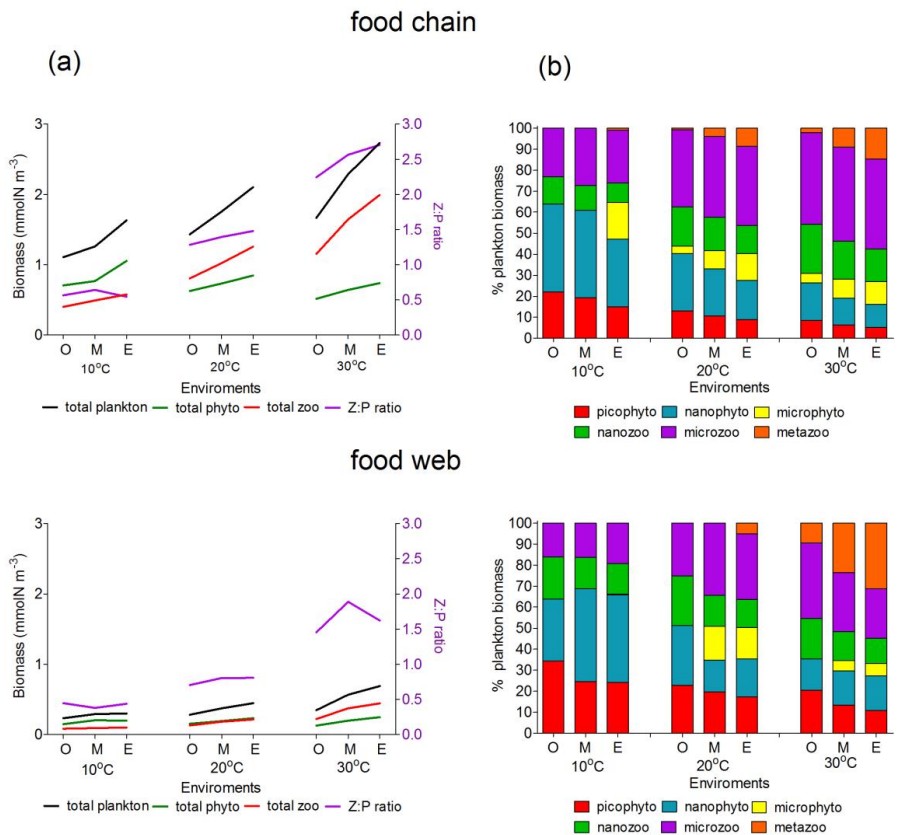


**Figure B1**. Plankton total biomass and group diversity for all environments (O: Oligotrophic, M: Mesotrophic and E: eutrophic environments). **(a)**: Right axis: biomass of phyto- (green line), zoo (red line) and total plankton (black line) (mmolNm⁻³). Left axis: zooplankton:phytoplankton biomass ratio (purple line). **(b)**: relative (%) biomass of phytoplankton and zooplankton size groups.






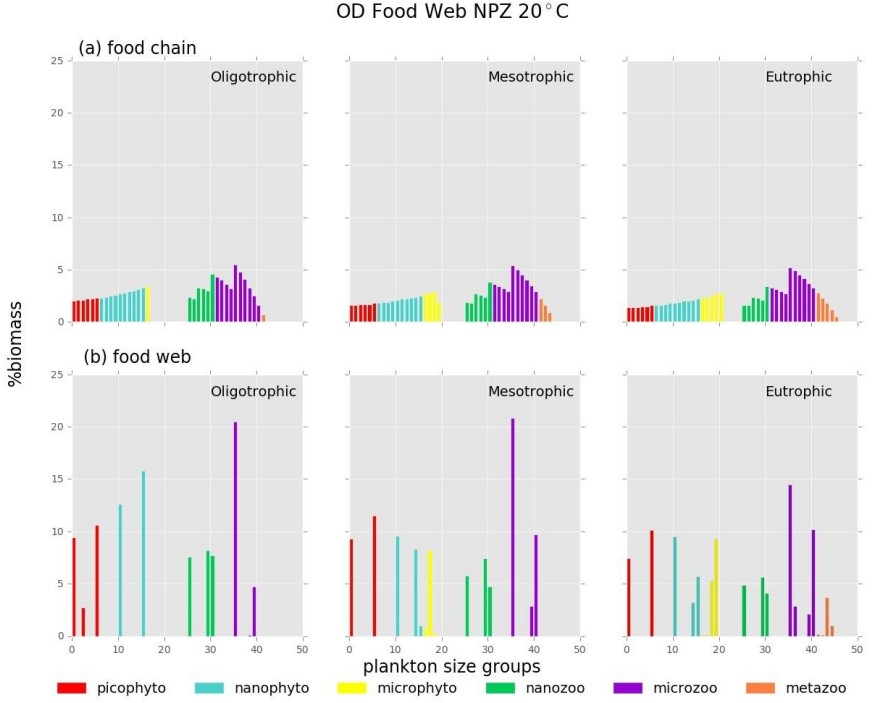

**Figure B2.** Relative biomass (%) of each phyto- and zooplankton group in (a) food chain and (b) food web for oligo-, meso- and eutrophic environments at 20ºC.





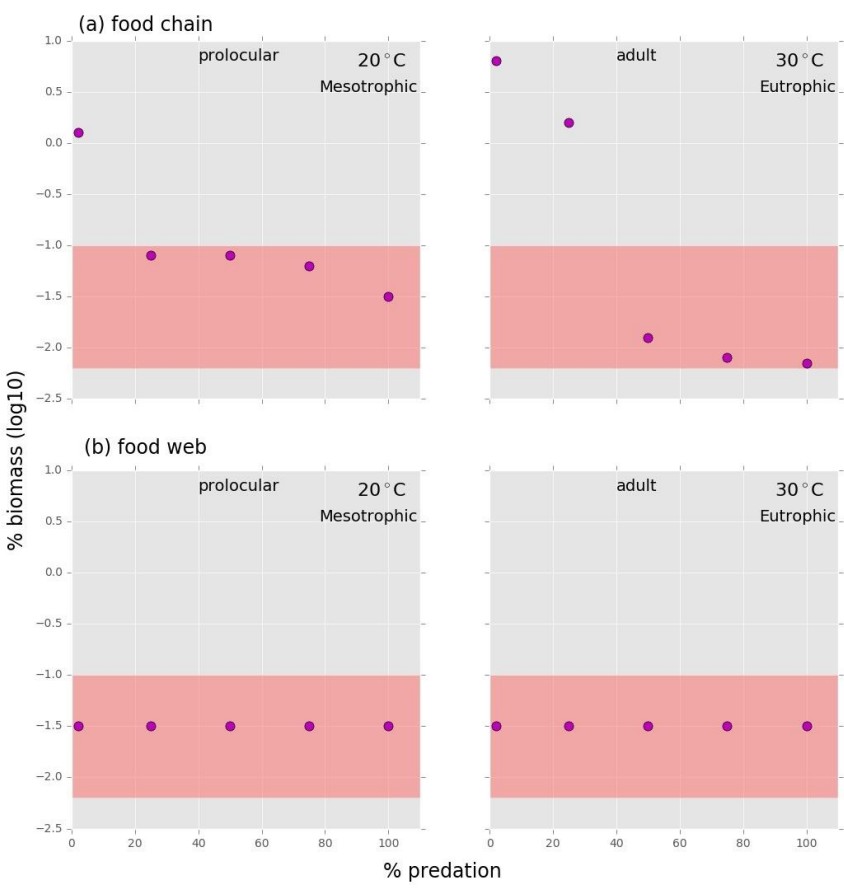

**Figure B3**: Results from the (a) food chain and (b) food web for different predation on planktonic foraminifera. With the coloured frame are the different grazing pressures on planktonic foraminifera for which their relative biomass is within the defined range (0.007% to 0.09%).
