# Peer review of "A trait-based modelling approach to planktonic foraminifera ecology"

_Biogeosciences, 2018_

## Referee Comment (RC1) · Anonymous Referee #1 · 1 Jan 2019

**1 General comments**

The authors present an impressive effort in developing and analysing the first trait-based model of planktonic foraminifera based on an existing size-based plankton model. As observations on foraminifera traits and trade-offs related to calcification are scarce, they use their model results to estimate costs and benefits by selecting plausible simulations from a large range of sensitivity simulations. They employ two different model trophic structures and reveal distinct effects of temperature and resource competition on prolocular and adult stages.

While the results are relevant, interesting and stimulate further research, the presentation of the model and the methods needs to be clarified to allow the reader to better

assess the predictive capability of the model and the generality of the model results. In particular, I would appreciate more details and clarification about the implications and effects of the grazing parameterisation described in Appendix B (see the following questions):

1. Could you please supply a plot of the different functional responses (with and withou prey refuge) employed?

2. You parameterised foraminifera as *the only group with no prey refuge to account for the cost of their inability to escape predation* (lines 899-900). Does the prey refuge operate only at low prey concentration or does it reduce predation rates at all prey concentrations? I guess an advantage due to escape ability should apply to any predator-prey encounter independent of prey concentration.

3. I would think that a prey refuge is a property of the predator and depends on the dominant foraging strategy in a given prey environment (Kiørboe et al. 2018), but does not differ for different prey types in a mixed environment. If the non-motile foraminifera were the only food available, would predators not likely choose an active feeding strategy exhibiting a feeding threshold (i.e. a prey refuge for the forams) to encounter their prey? In a mixed prey environment, I think it is not understood yet whether active or passive feeding strategy would be picked; but I find it hard to imagine that predators would switch to passive feeding without threshold/prey refuge when it perceives a foram (which it could not perceive well when ambushing anyways since the foram is non-motile (Kiørboe et al. 2010, Greve et al. 2017). In the absence of conclusive experimental evidence, I would find it reasonable to assume a prey refuge depending on total available food, but not differing for different types of prey and chose a different way to resolve potential escape reactions.

4. Does the applied formulation imply that at low food concentration foraminifera become the only zooplankton prey source for zooplankton? Could you describe if and how your results may change if the prey refuge is also applied to foraminifera?

5. You reduce the background mortality to allow foraminifera to achieve a high enough biomass to meet the 'low biomass' criterion, if I understood correctly (l238-240). Would it still need to be decreased if you allowed a prey refuge also for foraminifera?

6. As non-spinose foraminifera are immotile, I would expect them to be perceived and encountered at a similar rate as (immotile) phytoplankton of the same size (Visser 2007, Gonçalves and Kiørboe 2015). Motile zooplankton of similar size might be perceived easier and thus be encountered at higher frequency (Kiørboe et al. 2010, Almeda et al. 2017, Greve et al. 2017). Does your model resolve such a difference? If not, might this combined with the missing prey refuge add to the need of a reduced background mortality to allow foraminifera to coexist?

7. in line 904 you state that zooplankton are allowed to switch feeding behaviour from filter herbivorous to ambush carnivorous. Can you clarify in words under which conditions the zooplankton switch in your model? Do prolocular foraminifera as pure herbivores (cf. l.97) also switch? Do zooplankton in both model configurations (food chain and food web) switch?

References:

Almeda, R., H. v. S. Greve, and T. Kiørboe. 2017. Behavior is a major determinant of predation risk in zooplankton. Ecosphere 8. doi:10.1002/ecs2.1668.

Gonçalves, R. J., and T. Kiørboe. 2015. Perceiving the algae: How feeding-current feeding copepods detect their nonmotile prey. Limnol. Oceanogr. 60: 1286–1297. doi:10.1002/lno.10102.

[Figure]

Greve, H. v. S., R. Almeda, and T. Kiørboe. 2017. Motile behavior and predation risk in planktonic copepods. Limnol. Oceanogr. 62: 1810–1824. doi:10.1002/lno.10535.

Kiørboe, T., H. Jiang, and S. P. Colin. 2010. Danger of zooplankton feeding: the fluid signal generated by ambush-feeding copepods. Proc. R. Soc. Lond. B 277: 3229–3237. doi:10.1098/rspb.2010.0629.

Kiørboe, T., E. Saiz, P. Tiselius, and K. H. Andersen. 2018. Adaptive feeding behavior and functional responses in zooplankton. Limnology and Oceanography 63: 308–321. doi:10.1002/lno.10632.

Visser, A. W. 2007. Motility of zooplankton: fitness, foraging and predation. J. Plankton Res. 29: 447–461.

1. Does the paper address relevant scientific questions within the scope of BG?
   yes.

2. Does the paper present novel concepts, ideas, tools, or data?
   yes.

3. Are substantial conclusions reached?
   yes.

4. Are the scientific methods and assumptions valid and clearly outlined?
   in general yes, but the methods should be described more clearly.

5. Are the results sufficient to support the interpretations and conclusions?
   yes, but the above effects of the grazing parameterisation should be described/discussed.

6. Is the description of experiments and calculations sufficiently complete and precise to allow their reproduction by fellow scientists (traceability of results)?

partly. Some typos and missing values in equations/tables and some unclear wording might hinder reproduction.

7. Do the authors give proper credit to related work and clearly indicate their own new/original contribution?
yes.

8. Does the title clearly reflect the contents of the paper?
yes.

9. Does the abstract provide a concise and complete summary?
yes.

10. Is the overall presentation well structured and clear?
partly. The methods section should be improved and the wording in places.

11. Is the language fluent and precise?
partly. Some wording is misleading/confusing.

12. Are mathematical formulae, symbols, abbreviations, and units correctly defined and used?
mostly (few exceptions).

13. Should any parts of the paper (text, formulae, figures, tables) be clarified, reduced, combined, or eliminated?
the methods section and some figure legends (detailed below).

14. Are the number and quality of references appropriate?
yes.

15. Is the amount and quality of supplementary material appropriate?
yes.

**2 Specific comments**

1. l51, l217: I could not find the reference for Buitenhuis et al. 2014.

2. l57-59: unclear, could you reword?

3. l69: can you specify if the compilation of foraminifera traits was done as part of this study or otherwise add a reference ?

4. l71: what are "growth optimal environmental conditions" - environmental conditions that are optimal for growth?

5. l186: how do you mean more realistic: in terms of the setup or in terms of the results, or both? Productive/eutrophic regions are thought to have shorter food chain length than oligotrophic regions (**??**).

6. section 2.3: I find this section somewhat difficult to understand, partly because of typos and unclear wording. Would you make it clearer?

   - l216-217: is the Buitenhuis et al. 2014 data MAREDAT data for all micro- and mesozooplankton (there is only Buitenhuis et al. 2013 in the list of references) and the Schiebel and Movellan 2012 data for foraminifera?
   - l227: is the observed relative biomass range starting at 0.07%, 0.01% (l223) or 0.007% (l966)?
   - l227: are referred to here as 'low biomass' simulations.
   - l238: you need to decrease m to prevent foraminifera from going extinct, right? change to: "to keep planktonic foraminifera biomass within the 'low biomass' range defined above"
   - l240: this interpretation is difficult to follow at this point of the ms without having seen the results.

- l248: do you mean: "... to compare our ... results with, from the 'low biomass' simulations we selected model simulations with 0% to 30% reduction in maximum growth and background mortality ..."?
- l248: 0% to 30% reduction in [...] background mortality: in Figs. 4-7 and in the supplement xls file it looks like plausible simulations have 0% to 30% reduction in max growth rate but 0% to 50% reduction in beckground mortality. Also, the red font colour in the xls file is not clear: it seems to marks some, but not all simulations named LB.

7. section 2.4: can you describe more clearly which experiments were done in the text? Improving the layout of Table 3 (including legends for O, M and E) would also help.

8. l259-266: this section is very helpful information. Can you bring it earlier on in the methods section?

9. l290: you refer to steady state changes in Fig. B3 but "dynamics" sounds more like changes in time. How about "community structure" or "size structure"?

10. l314: "showed a decrease" instead of "resulted"?

11. l327-328: no 'plausible' simulations for the eutrophic ecosystem at 20 degC: but the bottom centre panel of Fig. 7 has a green star?

12. l476: does this refer to adults in the food chain model and prolocular stages in the food web model (cf. Fig. 5, 6), but not to adults in the food web model and prolocular stages in the food chain model?

13. Fig. 2: for which stage are the two schemes? As adults and prolocular stages have different size, I guess they would be positioned at different locations on the vertical/size "axis"?

14. Figs. 4-7: "total" in the legend is confusing. use "other"?; can you explain why the food chain simulations are all confined within an ellipse (Figs. 4, 5) while the food web shows this pattern only for some environmental conditions?

15. Eq. A1: what is R?

16. l890: the reference is missing in the list of references

**3  Technical corrections**

- l31: protection other than predation instead of "among"

- l54: either "to grow" or "to be grown"

- l61: can address (missing space)

- l86: responses instead of "responds"

- l96: feeders (missing s)

- l100: larger than themselves

- l121: Fraile et al. 2009? (missing e)

- l125: limitation is that they are based

- l126: from laboratory studies

- tense: please check uniform tense (past or present) particularly in sections 2.2, 2.3 (e.g. l200: assume instead of assumed, l203: is instead of was, ...)

- l216-222, l269: add spaces to units, e.g. Pg C instead of PgC

- l224: Schiebel and Movellan's (2012)

- l249: denoted as

- l280: the supplement has only figures B1-B3?

- l305: remove "the" before "all environments"

- l316: for the mesotrophic environment

- l317: due to a high decrease; compared with

- l394: ellipse?

- l416, 418: pellets

- l428: shells

- l431: it is diffficult

- l438: of grazing protection

- l445: controls

- l451: generalist herbivory and omnivory?

- l460-461: are very successful ... predators

- l476: oligotrophic

- l487: suggests

- l490: environment

- uniform referencing style for figures, equations: currently Fig., fig., figures, Eq, Eqs. eq() are all used.

- References list: please check carefully for typos (and citation format)

  - taxon names in italics (ll 570, 574, 576, 585)
  - l548: Foraminifera
  - l612: non-italic (d'Orbigny)
  - l620: Verlag
  - l643: trade-offs in
  - l659: Menden-Deuer
  - l660: . or , before journal name?

- l667: M $uren$

- Figs. 8, 9: Temperature

- Fig. 9: this is for the adult, not for the prolocular stage, right? please explain the different colour shading for the prey palatability

- l827: sizes

- l848: Eq. S2, S3 are A2, A3?

- l852: via the Monod

- l857: half-saturation

- l861: reflects instead of is represented

- l862: Prochlorococcus in italics

- l863: for the remaining (not rest)

- l876, 902: is $\Phi_{PVZ}$ the same as $\Phi_{PorZ}$?

- l889: on the Redfield; which ratio do you use?

- l892: K in italics

- l893: using Mayzaud

- l899: account for the

- l900: Kiørboe

- l905: ) as a; size (Gentleman

- l965: Within the coloured

Figure and Table layout:

- Tables 1-2: enhance clarity with the first column left justified and text on one line if possible

- Figs. 4-7: maybe use shading to identify the 'plausible' range of simulations?

- Figs. 8, 9: please mention the size ranges for pico, nano, micro here again

---

## Referee Comment (RC2) · Ayata (Referee) · 10 Jan 2019

Referee comment on the manuscript "A trait-based modelling approach to planktonic foraminifera ecology", by Maria Grigoratou and colleagues.

Referee: Sakina-Dorothée Ayata (Sorbonne Université, Laboratoire d'Océanographie de Villefranche sur mer, France).

**1) General comments**

This article present a new trait-based model for planktonic non-spinose foraminifera in order to test several trade offs among foraminifera feeding, growth and survival, and more specifically among size, trophic regime, feeding behaviour, predation avoidance,

and shell calcification.

The introduction is easy to follow and presents clearly all the needed information on planktonic foraminifera. However, the sentences on trait-based approaches could be rewritten to avoid some fuzziness in the presentation of the concept of trait. For instance, traits are defined at the individual level (see the recent review by Kiorboe, Visser Andersen, 2018, A trait-based approach to ocean ecology. ICES Journal of Marine Science (2018), doi:10.1093/icesjms/fsy090). The context of the study is clearly stated (model for foraminifera growth) and the study is well justified (need of a trait-based generic model, using body size, calcification, and feeding behaviour).

In the method section, the authors present the trait-based model of planktonic non-spinose foraminifera growth (including two life stages: prolocular and adult) they have developed in order to investigate the cost and benefits (trade-offs) of calcification and feeding behaviours under different environmental conditions (temperature and nutrient concentration).

The model set up adopted in the study is original and provides very interesting results.

The discussion is clear and relatively short, but the authors suggest several hypotheses to explain their results (observed trade-offs among calcification and growth) and the adequate literature is cited. Out of curiosity, I am wondering what type of trade-offs could exist in spinose foraminifera species.

The conclusion is clear and concise.

Therefore I recommend minor revisions before publication. Indeed, the presentation of the manuscript should be improved in order to present the model more clearly (see comments bellow).

**Review items**

1. Does the paper address relevant scientific questions within the scope of BG?

   Yes.

2. Does the paper present novel concepts, ideas, tools, or data?

   Yes (novel model).

3. Are substantial conclusions reached?

   Yes.

4. Are the scientific methods and assumptions valid and clearly outlined?

   The methods used should be outlined more clearly. See bellow.

5. Are the results sufficient to support the interpretations and conclusions?

   Yes.

6. Is the description of experiments and calculations sufficiently complete and precise to allow their reproduction by fellow scientists (traceability of results)?

   The code of the model is freely available.

7. Do the authors give proper credit to related work and clearly indicate their own new/original contribution?

   Yes.

   Minor comments: A recent review by Kiorboe et al 2018 on trait-based marine ecology could be added and the legend of Figure 1 should cite Litchman et al. 2013.

8. Does the title clearly reflect the contents of the paper?

   Yes.

9. Does the abstract provide a concise and complete summary?

   Yes.

10. Is the overall presentation well structured and clear?

    Yes, it is well structured, but the clarity could be improved (again, see bellow).

11. Is the language fluent and precise?

    Yes, although proof editing could prove useful.

12. Are mathematical formulae, symbols, abbreviations, and units correctly defined and used?

    Not clear enough in the present version.

13. Should any parts of the paper (text, formulae, figures, tables) be clarified, reduced, combined, or eliminated?

    Yes, clarifications are needed in the methods sections, in Table 3, and in the captions of the Figures. See bellow.

14. Are the number and quality of references appropriate?

    Yes.

15. Is the amount and quality of supplementary material appropriate?

    Yes.

**2) Specific comments**

**2.1) Remarks on the Model:**

I have several comments on the way the equations are indicated. Indeed, this section was difficult to follow, as many precious information was available in the annex and not in the main text of the manuscript.

I recommend to follow, when possible, writing standards for model equations (e.g., keep capital letters for variables and lower case letters for parameters, use mu for growth rates, etc. See comments bellow). The authors should make the equations much more clear, even for modellers, as this sections is difficult to follow.

Note that I appreciate that the code is freely available. I thank the authors for this effort of sharing their work to the scientific community.

**Miscellaneous comments on the model:**

l 116: what is a species model?

l 145: 2.1. Model environment => is "environment" the good term?

l 150: why is the duplication rate called kappa? Usually, it is called d or D in chemostat models.

l 146-147: Looking at your equations, it rather seems that nutrient availability is named N on your equations. Accordingly, replace the notation NO3- by N.

l153: each term of Equation 1 needs to be defined: what are $j_{prey}$, $J$, $B_{N,j}$, $P_{growth,j}$? Why $j_{prey}$ and not $j_{phyto}$ (indeed, zooplankton can be a prey, but would not do photosynthesis and impact the nutrient concentration)? Why do you use [ and ] in your equation? It seems not useful and hence confusing. Besides, it is usually written "parameter.Variable" in such differential equations: please reverse the writing and indicate: $p_{growth,j}.B_{N,j}$. More generally, please distinguish more clearly among parameters (lower case), functions (with brackets indicating their variables), and variables (capital letters).

Clearly indicate in the text that P and G are in fact functions and refer to the annex section.

l 156 to 159: move these sentences after having presented the equations with mortality terms and sloppy feeding terms.

l 163: shouldn't it be $B_{N,j}$ of $B_j$ rather than $B$ in the left side of equation 2? Note that the subscripts N are not useful here, unless you will later use an other currency than N for the biomass? Please rather indicate 3 equations: one for the autotrophic plankton, one for the heterotrophic plankton, and one for the mixotrophic plankton. What is $b$ in $\lambda_{i_{b,j}}$?

Impact of linear growth (instead of a Michaelis-Menten functional response) on your results? Considering only linear growth is a string assumption and the reasons to do so (and potential consequences) should be clearly indicated.

l 181: from a biological and ecological point of view, what would be a "specialist predator on planktonic foraminifera", as included in your simple food chain model? What would be its characteristics?

l 222: is it realistic to consider that the "protocolar biomass is similar to the adult biomass"? I would have expected to have much more protocolar biomass than adult biomass, especially given their slow growth (but I am not a specialist of foraminifera...)

l 260: Table 3 is difficult to follow as the horizontal and vertical lines are not indicated. Please make it easier to read. For instance, why is there 3 identical rows for Nutrient region? I would assume that you used the 3 different regimes (O, M, E) for each of the Temperature conditions (10, 20, 30), but it is not what I read in Table 3. Similarly, in the part entitled "Study traits", the rows of "Prolocular (20 $\mu$m)" and "Adult (160 $\mu$m)" are identical. If this is correct, then please merge them.

**2.2) Remarks on the Results:**

- I find it strange to start this section with Figures that are all in Annex and not in the main text (Figures B1 and B2).

**2.3) Remarks on the Figures:**

Fig. 1: useful, but indicated in the legend that this figure is inspired from the topology of zooplankton traits proposed by Litchman et al 2013 in JPR.

Legend of Figure 3: indicate the name of the parameter $\sigma$.

Figures 4 to 7: the symbols for 'plausible' and 'low biomass' are very difficult to distinguish, especially because the stars and triangles are light green on a light grey background. Please modify (an provide figures with a better resolution).

Figure 4 to 7: why not use the same setting as in Figures 8 to 9, with an horizontal arrow indicating the increase in Temperature (please correct the typo: Tempertature), and a vertical arrow indicated the increase in Nutrient concentration (O-M-E) ?

**3) Technical corrections:**

Please find bellow additional minor comments:

- Check and remove double spaces throughout the ms

- Please revise the manuscript to remove all typos, for instance (in the beginning of the manuscript, I have not indicated all of them here):

l 23: extra space (on trait theory)

l 56: change in police size

l 61: no space (canadress)

l 90-91 : change in police size

l 103: " It has been speculated that the higher abundance...": higher than what?

l 125: the subject ("they"?) is missing in "is that are based"

l 139: no comma in "interactions of planktonic foraminifera, with..."

---

## Author Comment (AC1) · 15 Feb 2019

We would like to thank the reviewer for her strongly supportive comments towards our study as well as her thoughtful and constructive points. The latter helped us to improve the description of our model in the method section and Appendix of our manuscript. Please find below our reply to reviewer's comments (highlighted in blue).

5

**Referee 2**

This article presents a new trait-based model for planktonic non-spinose foraminifera in order to test several trade-offs among foraminifera feeding, growth and survival, and more specifically among size, trophic regime, feeding behaviour, predation avoidance, and shell calcification.

- 10 size, trophic regime, feeding behaviour, predation avoidance, and shell calcification. The introduction is easy to follow and presents clearly all the needed information on planktonic foraminifera. However, the sentences on trait-based approaches could be rewritten to avoid some fuzziness in the presentation of the concept of trait. For instance, traits are defined at the individual level (see the recent review by Kiorboe, Visser Andersen, 2018, A trait-based approach to ocean
- 15 ecology. ICES Journal of Marine Science (2018), doi:10.1093/icesjms/fsy090). The context of the study is clearly stated (model for foraminifera growth) and the study is well justified (need of a traitbased generic model, using body size, calcification, and feeding behaviour).

**We change the presentation of traits to: "Trait-based approaches provide mechanistic understanding of individuals, populations or ecosystems functioning as they describe these systems from first principles by defining individuals' key traits (e.g. feeding, competition, predation, reproduction) and associated trade-offs like energetic needs and predation risks (e.g. Litchman and Klausmeier, 2008; Litchman et al., 2013; Barton et al., 2016; Hébert et al., 2016; Kiørboe, 2018)." (163-65)**

25

In the method section, the authors present the trait-based model of planktonic nonspinose foraminifera growth (including two life stages: prolocular and adult) they have developed in order to investigate the cost and benefits (trade-offs) of calcification and feeding behaviours under different environmental conditions (temperature and nutrient concentration). The model set up adopted in the study is original and provides very interesting results.

30 the study is original and provides very interesting results. The discussion is clear and relatively short, but the authors suggest several hypotheses to explain their results (observed trade-offs among calcification and growth) and the adequate literature is cited. Out of curiosity, I am wondering what type of trade-offs could exist in spinose foraminifera species.

35

We would like to thank the reviewer for finding out discussion clear and useful. A more detailed review study on planktonic foraminifera (non-spinose and spinose) traits and trade- offs is in preparation from Edgar, Monteiro, Grigoratou and Schmidt.

40 The conclusion is clear and concise. Therefore, I recommend minor revisions before publication. Indeed, the presentation of the manuscript should be improved in order to present the model more clearly (see comments below).

**2) Specific comments**

**45 **2.1) Remarks on the Model:**

I have several comments on the way the equations are indicated. Indeed, this section was difficult to follow, as many precious information was available in the annex and not in the main text of the manuscript.

I recommend to follow, when possible, writing standards for model equations (e.g., keep capital letters for variables and lower case letters for parameters, use mu for growth rates, etc. See

comments bellow). The authors should make the equations much more clear, even for modellers, as this sections is difficult to follow.

Note that I appreciate that the code is freely available. I thank the authors for this effort of sharing their work to the scientific community.

55

50

**Miscellaneous comments on the model:**

I 116: what is a species model?

60 We apologise for the confusion, we meant models which are built for specific species of planktonic foraminifera. We rephrased our sentence to *"However, until now, only species-specific ecological models* have been developed to study the ecology of modern planktonic foraminifera species: Žarić et al. (2006) (from now on Žarić06), PLAFOM (Fraile et al., 2008; Fraile et al., 2009) and FORAMCLIM (Lombard et al., 2011; Roy et al., 2015)." (I115)

65

I 145: 2.1. Model environment => is "environment" the good term? We changed model environment to model description (I145).

l 150: why is the duplication rate called kappa? Usually, it is called d or D in chemostat models.

70 As we built our model from Ward et al. (2012) and Ward et al. (2014) modelling studies, we followed their symbols and definitions for consistency.

l 146-147: Looking at your equations, it rather seems that nutrient availability is named N on your equations. Accordingly, replace the notation NO3- by N.

75 Done.

I153: each term of Equation 1 needs to be defined: what are  $j_{prey}$ ,  $B_{N,j}$ ,  $P_{growth,j}$ ? Why  $j_{prey}$  and not  $j_{phyto}$  (indeed, zooplankton can be a prey, but would not do photosynthesis and impact the nutrient concentration)? Why do you use [ and ] in your equation? It seems not useful and hence confusing. Besides, it is usually written "parameter. Variable" in such differential equations: please reverse the writing and indicate:  $P_{growth,j}B_{N,j}$ . More generally, please distinguish more clearly among parameters (lower case), functions (with brackets indicating their variables), and variables

(capital letters). Clearly indicate in the text that P and G are in fact functions and refer to the annex

85

90

section.

80

We present the equations in a similar format to Ward et al., 2012, for consistency. We added the following sentences to the manuscript (l166-169) "Phytoplankton growth ( $P_{growth,j}$ ) depends on limitation from light, temperature and nutrient availability, following a Monod response (Appendix, eq. A2). Zooplankton grazing is controlled by the biomass and size of the prey and is described through a Holling type II response (Eq. (A3))."

I 156 to 159: move these sentences after having presented the equations with mortality terms and sloppy feeding terms.

95 Done.

l 163: shouldn't it be BN; j of Bj rather than B in the left side of equation 2? Note that the subscripts N are not useful here, unless you will later use an other currency than N for the biomass? Please rather

indicate 3 equations: one for the autotrophic plankton, one for the heterotrophic plankton, and one for the mixotrophic plankton. What is b in  $\lambda_{ib,i}$ ?

- for the mixotrophic plankton. What is b in λib,j?
  We have removed the "N" from the biomass and the b in λ, which was a typographic mistake. In our model we include only autotrophs and heterotrophs but no mixotrophs. For consistency with Ward et a., 2012, we decide to keep one generic equation for plankton biomass (Eq.2).
- 105 Impact of linear growth (instead of a Michaelis-Menten functional response) on your results? Considering only linear growth is a string assumption and the reasons to do so (and potential consequences) should be clearly indicated.

As specified above, the growth of phytoplankton depends on the limitations from light, temperature and nutrient availability, which follow a Monod-type response (Appendix, eq. (A2))

110

1181: from a biological and ecological point of view, what would be a "specialist predator on planktonic foraminifera", as included in your simple food chain model? What would be its characteristics?

- 115 The prey-predator interactions between planktonic foraminifera and other zooplankters are still not well understood. Parts of planktonic foraminifera's shells have been found on salps faecal pellets, which are filter feeders and hence non-specialised. Current evidence suggests that planktonic foraminifera do not have a specific zooplankton predator and they are indiscriminately grazed by filter feeding organisms (Hemleben et al., 1989). As one aim of our study is to understand the role of
- 120 shell as protection from predation, we chose to test two scenarios, one where predators are specialised on planktonic foraminifera (i.e. planktonic foraminifera are the only zooplankton group which prey on, food chain), and one where opportunist predators can use planktonic foraminifera as part of their diet (food web). From a biological and theoretical point of view having a specialist predator on plankton foraminifera seems to be unrealistic but as predation on planktonic foraminifera is not well understood we chose to test both hypotheses.

Hemleben, C., Spindler, M. and Anderson, O.R.: Modern Planktonic Foraminifera. Springer Verlag, New York, 1989, p135.

130

I 222: is it realistic to consider that the "protocolar biomass is similar to the adult biomass"? I would have expected to have much more protocolar biomass than adult biomass, especially given their slow growth (but I am not a specialist of foraminifera...)

- 135 This is a fair point. It has been suggested that the biomass of early stages can be up to three times higher than adults (Schiebel and Movellan, 2012) but data regarding the abundance and biomass of planktonic foraminifera's early stages are scare due to sample limitations specifically a combination of their planktonic foraminifera's low abundance and the focus on nets with mesh size >100 µm. For the present study, we decided to extend our estimations for the size fraction 150-200 µm and
- 140 converted the biomass range by a factor of three to include a global representation of planktonic foraminifera, sampling errors, and early stages. As planktonic foraminifera's cytoplasm (organic biomass) is growing parallel with the shell, we argue that our estimated biomass range for the adult with size 160µm can also represent the early stages, even though prolocular's abundance is higher than the adults, adult's biomass per individual is higher due to their bigger cytoplasm. Based on that
- 145 we believe that the prolocular biomass cannot be less than define range. More biomass data are needed in order to improve our biomass estimations, but we believe that as the aim of our study is

to explore the costs and benefits of calcification in different life stages, our biomass range can be used for both life stages.

- 150 Buitenhuis, E. M., Vogt, R., Moriarty, N., Bednarsek, S.C., Doney, S. C., Leblanc, K., Le Quéré, C., Luo, Y. W., O'Brien, C., O'Brien T., Peloquin J., Schiebel, R., C. Swan, C.: MAREDAT: towards a world atlas of MARine Ecosystem DATa. Earth System Science Data, Copernicus Publications, 5, 227-239 https://doi.org/10.5194/essd-5-227-2013, 2013.
- 155 Schiebel, R. and Movellan, A.: First-order estimate of the planktic foraminifer biomass in the modern ocean, Earth Syst. Sci. Data, 4, 75-89, https://doi.org/10.5194/essd-4-75-2012, 2012.

I 260: Table 3 is difficult to follow as the horizontal and vertical lines are not indicated. Please make
 it easier to read. For instance, why is there 3 identical rows for Nutrient region? I would assume that you used the 3 different regimes (O, M, E) for each of the Temperature conditions (10, 20, 30), but it is not what I read in Table 3. Similarly, in the part entitled "Study traits", the rows of "Prolocular (20 m)" and "Adult (160 m)" are identical. If this is correct, then please merge them.

Thank you for these useful suggestions. We applied all changes to our Table.

165

185

**2.2) Remarks on the Results:**

- I find it strange to start this section with Figures that are all in Annex and not in the main text (Figures B1 and B2).

170 We added Figure B1 into the main text (now referred as Fig.3) and left Figure B2 in the Appendix A.

**2.3) Remarks on the Figures:**

Fig. 1: useful, but indicated in the legend that this figure is inspired from the topology of zooplankton traits proposed by Litchman et al 2013 in JPR.

175 We added the following sentence into the legend: *"The presentation of planktonic foraminifera's traits was inspired from the topology of zooplankton traits proposed by Litchman et al. (2013)."* (1789-790).

Legend of Figure 3: indicate the name of the parameter  $\sigma$ .

180 We now included the figure of prey palatability in Fig.2 and indicated the name of parameter ( $\sigma$ ) (1800).

Figures 4 to 7: the symbols for 'plausible' and 'low biomass' are very difficult to distinguish, especially because the stars and triangles are light green on a light grey background. Please modify (an provide figures with a better resolution).

We changed the colour scale and added black edges for the 'plausible' and 'low biomass' symbols to make the figures easier to read.

Figure 4 to 7: why not use the same setting as in Figures 8 to 9, with an horizontal arrow indicating the increase in Temperature (please correct the typo: Tempertature), and a vertical arrow indicated the increase in Nutrient concentration (O-M-E) ? Done.

**195 3) Technical corrections:**

Please find bellow additional minor comments: Done

- Check and remove double spaces throughout the ms Done
- Please revise the manuscript to remove all typos, for instance (in the beginning of the
- manuscript, I have not indicated all of them here):
- 200 l 23: extra space (on trait theory). Done
  - l 56: change in police size. Done
    - l 61: no space (canadress). Done
    - I 90-91 : change in police size. Done
- I 103: "It has been speculated that the higher abundance...": higher than what?. We changed that to
  *"higher abundance of spinose species compared to the non-spinose is the result of their carnivory"* (1104).
  - I 125: the subject ("they"?) is missing in "is that are based". Done
  - I 139: no comma in "interactions of planktonic foraminifera, with...". Done

---

## Author Comment (AC2) · 15 Feb 2019

We would like to thank the reviewer for the strongly supportive comments towards our study as well as the thoughtful and constructive points. The latter helped us to improve the description of our model in the method section and Appendix of our manuscript. Please find below our reply to each reviewer's comments (highlighted in blue).

Referee 1

The authors present an impressive effort in developing and analysing the first trait-based model of planktonic foraminifera based on an existing size-based plankton model. As observations on foraminifera traits and trade-offs related to calcification are scarce, they use their model results to estimate costs and benefits by selecting plausible simulations from a large range of sensitivity simulations. They employ two different model trophic structures and reveal distinct effects of temperature and resource competition on prolocular and adult stages. While the results are relevant, interesting and stimulate further research, the presentation of the model and the methods needs to be clarified to allow the reader to better assess the predictive capability of the model and the generality of the model results. In particular, I would appreciate more details and clarification about the implications and effects of the grazing parameterisation described in Appendix B (see the following questions):

1. Could you please supply a plot of the different functional responses (with and without prey refuge) employed?

We have added in the Appendix A a plot showing the different grazing rate with and without a prey refuge (Figure A1, also shown below as Figure 1) as well as including a better description of the prey refuge term (l945-979). The prey refuge in our model accounts for the fact that a prey at a low biomass is hard to find, and thus less prone to predation. Therefore, when the prey refuge is included in the grazing term, the grazing rate decreases with low densities (Figure A1).

[Figure]

Figure A1: Zooplankton grazing on one prey with and without the prey refuge term included. Prey refuge $= (1 - e^{-\Lambda F}) * F$. Grazing without prey refuge: $G = G_{max} * \gamma_T * \frac{F}{F + K_{zoo}}$. Grazing with prey refuge included : $G = G_{max} * \gamma_T * \frac{F}{F + K_{zoo}} *$ Prey refuge. Temperature limitation ($\gamma_T$), prey palatability ($\varphi$) and prey refuge constant ($\Lambda$) equals to 1, and $F = \varphi * B$.

2. You parameterised foraminifera as the only group *with no prey refuge to account for the cost of their inability to escape predation* (lines 899-900). Does the prey refuge operate only at low prey concentration or does it reduce predation rates at all prey concentrations? I guess an advantage due to escape ability should apply to any predator-prey encounter independent of prey concentration.

Thank you for this valid point. As explained above, the prey refuge term in our model relies on prey's density and palatability and decreases the predation rate when the prey's biomass is low. As a result, the prey refuge term in our model does not represent directly a motility effect as might be suggested here. In the real ocean, prey can use defence mechanisms to escape predation (including motility but not exclusively, e.g. toxins, shell, spines, colony formation), which are independent of their density. Our model does not include these mechanisms because of still too limited mechanistic understanding (e.g. Pančić and Kiørboe, 2018).

Pančić M., and Kiørboe, T.: Phytoplankton defence mechanisms: traits and trade-offs, *Biological reviews,* 93 pp. 1269 – 1303, https://doi.org/10.1111/brv.12395, 2018.

3. I would think that a prey refuge is a property of the predator and depends on the dominant foraging strategy in a given prey environment (Kiørboe et al. 2018) but does not differ for different prey types in a mixed environment. If the non-motile foraminifera were the only food available, would predators not likely choose an active feeding strategy exhibiting a feeding threshold (i.e. a prey refuge for the forams) to encounter their prey? In a mixed prey environment, I think it is not understood yet whether active or passive feeding strategy would be picked; but I find it hard to imagine that predators would switch to passive feeding without threshold/prey refuge when it perceives a foram (which it could not perceive well when ambushing anyways since the foram is non-motile (Kiørboe et al. 2010, Greve et al. 2017). In the absence of conclusive experimental evidence, I would find it reasonable to assume a prey refuge depending on total available food, but not differencing for different types of prey and chose a different way to resolve potential escape reactions.

The predator prey interactions depend mostly on predator-prey length ratio (Kiørboe, 2008), prey's availability (Kiørboe, 2008) and ability to escape predation (van Someren Gréve et al., 2017; Pančić and Kiørboe, 2018) and predator's feeding behaviour (Kiørboe et al., 2018). The prey refuge term in our model only includes prey's density and palatability terms as it has been described in our reply for comments 1 and 2. We chose to exclude the prey refuge term for predation on planktonic foraminifera in order to study better the role of planktonic foraminifera's shell as a protection against predation, assuming that the excluded prey refuge balances their immotility. We chose to increase the complexity of the model and the uncertainty of the results, by adding motility. Therefore, in the discussion we emphasize that including motility in the model is the next important step for better understanding the predation on planktonic foraminifera (l455-456). For more regarding planktonic foraminifera motility please read our reply to reviewer's comment 6.

Kiørboe, T.: A mechanistic approach to plankton ecology. Princeton University Press, p.107-114,2008.
Kiørboe, T., Saiz, E., Tiselius, P., and Andersen, K.H.: Adaptive feeding behaviour and functional responses in zooplankton. Limnology and Oceanography 63: 308–321. https://doi:10.1002/lno.10632, 2018.
van Someren Gréve, H., Almeda, R. and Kiørboe, T.: Motile behavior and predation risk in planktonic copepods, Limnology and Oceanography, https://doi.org/10.1002/lno.10535, 2017.

Pančić, M., and Kiørboe, T.: Phytoplankton defence mechanisms: traits and trade-offs, *Biological reviews,* 93 pp. 1269 – 1303, https://doi.org/10.1111/brv.12395, 2018.

4. Does the applied formulation imply that at low food concentration foraminifera become the only zooplankton prey source for zooplankton? Could you describe if and how your results may change if
the prey refuge is also applied to foraminifera?

Planktonic foraminifera in our "low biomass" and "plausible" scenarios exist with very low biomass and coexist with another zooplankton group of the same size. The latter has a higher biomass than the foraminifera so that foraminifera is never the dominant source of food. With our model, we tested
the effect of prey refuge on foraminifera ecology and found that for both versions of the model, the mortality rate should be decrease in order to keep planktonic foraminifera's biomass within the observation range with and without the prey refuge being included. We added this information on the Appendix A l974-980.

5. You reduce the background mortality to allow foraminifera to achieve a high enough biomass to meet the 'low biomass' criterion, if I understood correctly (l238-240). Would it still need to be decreased if you allowed a prey refuge also for foraminifera?

In both versions of our model (food chain and food web), we need to decrease the background
mortality in order to sustain planktonic foraminifera biomass within the observed range with or without the prey refuge term. This is thus a robust result as well which we refer too now as well in the manuscript (l974-980).

6. As non-spinose foraminifera are immotile, I would expect them to be perceived and encountered
at a similar rate as (immotile) phytoplankton of the same size (Visser 2007, Gonçalves and Kiørboe 2015). Motile zooplankton of similar size might be perceived easier and thus be encountered at higher frequency (Kiørboe et al. 2010, Almeda et al. 2017, Greve et al. 2017). Does your model resolve such a difference? If not, might this combined with the missing prey refuge add to the need of a reduced background mortality to allow foraminifera to coexist?

This is a very good observation. The model does not represent difference in motility within the plankton community. Despite being important, as pointed out here by the reviewer, no published study exists regarding planktonic foraminifera's motility. Based on our general understanding of motility on animal and plankton ecology (e.g. Broglio et al., 2001; Almeda et al., 2017), we suggest
that the main traits of planktonic foraminiferal immotility are to save energy and protect from predation. The associated trade-offs of immotility are lower prey efficiency and inability of escape from predation when they encounter their predator. As the focus of the present study is to understand the costs and benefits of calcification, we did not add motility as this would increase the complexity of the model and the uncertainty of the results in light of the lack of experimental evidence on
planktonic foraminifera. We chose to exclude the prey refuge to study better the function of the shell as an armour against predation by making the assumption that the lack of prey refuge could balance the cost of their immotility. We agree with the reviewer that by removing the prey refuge term is a very simple way to represent that and not the correct one. We believe that exploring the traits and trade-offs of planktonic foraminifera motile is an important next step for studying further the
predation of planktonic foraminifera; as we stated in the discussion (l454-462).

Almeda, R., van Someren Gréve, H., and Kiørboe, T.:. Behavior is a major determinant of predation risk in zooplankton. Ecosphere 8(2). https://doi:10.1002/ecs2.1668, 2017.

Broglio, E., Johansson, M. and Jonsson, P. R.: Trophic interaction between copepods and ciliates: effects of prey swimming behavior on predation risk. Marine Ecology Progress Series 179: 179– 186, https://doi.org/10.3354/meps220179, 2001.

7. in line 904 you state that zooplankton are allowed to switch feeding behaviour from filter herbivorous to ambush carnivorous. Can you clarify in words under which conditions the zooplankton switch in your model? Do prolocular foraminifera as pure herbivores (cf. l.97) also switch? Do zooplankton in both model configurations (food chain and food web) switch?

Thank you for pointing this out. We improved the description of zooplankton feeding in the manuscript (l973-977) as it was misleading. In the food web version of the model, zooplankton are defined as omnivorous predators, able to consume in parallel more than one phytoplankton and zooplankton preys but we do not distinguish them between herbivorous or carnivorous. The predator can actively choose to feed mostly on phytoplankton ($\Phi_P$) or zooplankton ($\Phi_Z$) prey, depending on prey's palatability ($\varphi_{j_{pred},j_{prey}}$) and density ($B_{j_{prey}}$) weighted in total prey density ($B_{prey}$) (Gentleman et al., 2003; Ward et al., 2012). For the food chain version of our model, zooplankton are defined as specialist herbivorous predators. As we wanted to test the benefit of shell protection from a specialist predator on planktonic foraminifera biomass, we made an exception by defining one zooplankton group to be omnivorous, feeding with one phytoplankton prey and planktonic foraminifera only. For both versions of our model, both life stages of planktonic foraminifera are defined as herbivorous feeders only.

Gentleman, W. C., Leising, A., Frost, B., Strom, S., Murray, J.: Functional responses for zooplankton feeding on multiple resources: A review of assumptions and biological dynamics, Deep-Sea Research II, 50: 2847–2875, https://doi.org/10.1016/j.dsr2.2003.07.001, 2003.

Ward, B. A., Dutkiewicz, S., Jahn, O., Follows, M.J.: A size-structured food-web model for the global ocean. Limnology Oceanography, 57(6), p.1877–1891, http://dx.doi.org/10.4319/lo.2012.57.6.1877, 2012.

**2 Specific comments**

1. l51, l217: I could not find the reference for Buitenhuis et al. 2014.

It is a mistake. We have now corrected this.

2. l57-59: unclear, could you reword?

We rephrased the sentence to: *"The development and application of numerical ecological models can help fill in this knowledge gap. Particularly promising to address ecological questions, **trait-based models examine individual's** physiological traits and their associated trade-offs." (l58-59)*

3. l69: can you specify if the compilation of foraminifera traits was done as part of this study or otherwise add a reference ?

This is the first time that an overview of the traits and trade- offs of planktonic foraminifera has been presented.  We are in the process of writing a more in-depth review study of planktonic foraminifera's traits and trade-offs to be submitted in the coming year (Edgar, Monteiro, Grigoratou and Schmidt, in prep).

4. l71: what are "growth optimal environmental conditions" - environmental conditions that are optimal for growth?

That is correct. We modified as suggested in the manuscript (l71).

5. l186: how do you mean more realistic: in terms of the setup or in terms of the results, or both? Productive/eutrophic regions are thought to have shorter food chain length than oligotrophic regions (??).

We meant that the food web is more realistic in terms of the setup. This is because it better represents the predator: prey interactions between phytoplankton and zooplankton communities than the food chain model. These trophic interactions are commonly found in both oligotrophic and eutrophic regions (e.g. Calbet and Saiz, 2005). We clarify this point now in the manuscript (l192-193).

Calbet, A. and Saiz, E: The ciliate-copepod link in marine ecosystems, Aquat Microb Ecol, 38: 157–167, 2005.

6. section 2.3: I find this section somewhat difficult to understand, partly because of typos and unclear wording. Would you make it clearer?

• l216-217: is the Buitenhuis et al. 2014 data MAREDAT data for all micro and mesozooplankton (there is only Buitenhuis et al. 2013 in the list of references) and the Schiebel and Movellan 2012 data for foraminifera?

That is correct. We improved the section "planktonic foraminifera biomass" in order to reduce confusion (l230-251).

• l227: is the observed relative biomass range starting at 0.07%, 0.01% (l223) or 0.007% (l966)?

The observed relative biomass range should be 0.007 % to 0.09 % (and not 0.07% as we wrote in l227). We corrected this error in l223 to l227.

• l227: are referred to here as 'low biomass' simulations.

Done.

• l238: you need to decrease m to prevent foraminifera from going extinct, right? change to: "to keep planktonic foraminifera biomass within the 'low biomass' range defined above"

Changed accordingly in l264.

• l240: this interpretation is difficult to follow at this point of the ms without having seen the results.

We rephrased the sentence to *"following suggestions that planktonic foraminifera can use their shell as a protection against other factors than predation (e.g. pathogens) (Armstrong and Brasier, 2005)."* (l264-265)

• l248: do you mean: "... to compare our ... results with, from the 'low biomass' simulations we selected model simulations with 0% to 30% reduction in maximum growth and background mortality ..."?

Until now there are no direct quantifying estimates of the energetic cost and benefits of calcification in planktonic foraminifera to compare our model results with. Therefore, we selected a maximum 40% range between the minimum and maximum value of maximum growth ($G_{max}$) and mortality (m) reduction of as most likely (e.g., 10-50 % or 20-60 % reduction). This is a way to account for the non-unlimited plasticity of an organism. We added this information in l272-276.

• l248: 0% to 30% reduction in [...] background mortality: in Figs. 4-7 and in the supplement xls file it looks like plausible simulations have 0% to 30% reduction in max growth rate but 0% to 50% reduction in background mortality. Also, the red font colour in the xls file is not clear: it seems to mark some, but not all simulations named LB.
The range between the minimum and maximum value of each growth and background mortality is indeed less than 40%. The red mark in the xls file represent the biomass within the observation range, which applies for all the 'plausible' and 'low biomass (LB)' simulations and some scenarios testing different predation rates. As this clarification wasn't included in the excel file, we agree that the red mark was confusing and have changed it accordingly. The following information has now been added in every excel sheet: *"In red: biomass within the observation range. "*

7. section 2.4: can you describe more clearly which experiments were done in the text? Improving the layout of Table 3 (including legends for O, M and E) would also help.
We improved the layout of Table 3 as suggested.

8. l259-266: this section is very helpful information. Can you bring it earlier on in the methods section?
This paragraph has been now transferred in the beginning of the 2.3 section (l221-227).

9. l290: you refer to steady state changes in Fig. B3 but "dynamics" sounds more like changes in time. How about "community structure" or "size structure"?
This is a very good suggestion. We rephrased the sentence ass suggested (l315)

10. l314: "showed a decrease" instead of "resulted"?
Done.

11. l327-328: no 'plausible' simulations for the eutrophic ecosystem at 20 °C: but the bottom centre panel of Fig. 7 has a green star?

That is correct. We do have a plausible scenario for the eutrophic ecosystem at 20°C. We have corrected this mistake in the manuscript (l355).

12. l476: does this refer to adults in the food chain model and prolocular stages in the food web model (cf. Fig. 5, 6), but not to adults in the food web model and prolocular stages in the food chain model?

We can see there could be a confusion and rephrased the sentence to clarify this point:
*"Moreover, the inability of our **food web** model to sustain **adult** stages of non-spinose foraminifera in warm oligotrophic regions agrees with observations as planktonic foraminifera are dominated by*

*symbiont bearing species in these regions (Bé and Tolderlund, 1971). "* (l500-503).

13. Fig. 2: for which stage are the two schemes? As adults and prolocular stages have different size, I guess they would be positioned at different locations on the vertical/size "axis"?
We agree that it might not be very clear from the plot in Figure 2. As suggested, the prolocular life stage should be at a lower position on plankton size "axis" than the adult one. To prevent this type of confusion, we added the following sentence to the figure legend:

*"Note that the figure does not present the accurate position of the planktonic foraminifera size group ran in the model but a generic position for illustrate how they interact with the rest of the plankton community."*

14. Figs. 4-7: "total" in the legend is confusing. use "other"?; can you explain why the food chain simulations are all confined within an ellipse (Figs. 4, 5) while the food web shows this pattern only for some environmental conditions?

We changed the "total" to "other" after the reviewer's useful suggestion. We tested fewer simulations in the food web model than in the food chain model, hence a different spread of model results in Figures 4 and 5. We tested fewer simulations based on our food chain model results that showed that in many tested scenarios, planktonic foraminifera were extinct. On the contrary, for the food web, in a range of a 0 to 40% reduction on the mortality rate, the relative biomass of planktonic foraminifera was high and outside the observation range. As a further reduction of the mortality rate would result in an additional increase of relative biomass, the sensitivity analysis was not required. We added the above explanation in Figures 6 and 7 (l835-839, l846-851).

15. Eq. A1: what is R?

R is temperature sensitivity of plankton growth rate. We defined R in the text (l894-895) and Table 1.

16. l890: the reference is missing in the list of references
The reference has now been included.

**3 Technical corrections**

• l31: protection other than predation instead of "among". Done
• l54: either "to grow" or "to be grown". Done
• l61: can address (missing space) . Done
• l86: responses instead of "responds". Done
• l96: feeders (missing s) . Done
• l100: larger than themselves. Done
• l121: Fraile et al. 2009? (missing e). Done
• l125: limitation is that they are based. Done
• l126: from laboratory studies. Done
• tense: please check uniform tense (past or present) particularly in sections 2.2, 2.3 (e.g. l200: assume instead of assumed, l203: is instead of was, ...). Done
• l216-222, l269: add spaces to units, e.g. Pg C instead of PgC. Done
• l224: Schiebel and Movellan's (2012). Done
• l249: denoted as. Done
• l280: the supplement has only figures B1-B3?
The figure B1-B3 are part of the appendix B. The supplement materials include the code scripts and the excel file with all the ran simulations.
• l305: remove "the" before "all environments". Done
• l316: for the mesotrophic environment. Done
• l317: due to a high decrease; compared with. Done
• l394: ellipse? Done
• l416, 418: pellets. Done
• l428: shells. Done

• l431: it is difficult. Done

• l438: of grazing protection. Done

• l445: controls. Done

• l451: generalist herbivory and omnivory? We change the sentence to *"generalist herbivory and omnivory diet"* (l476)

• l460-461: are very successful ... predators. Done

• l476: oligotrophic. Done

• l487: suggests. Done

• l490: environment. Done

• uniform referencing style for figures, equations: currently Fig., fig., figures, Eq, Eqs. eq() are all used.

Done

• References list: please check carefully for typos (and citation format)

– taxon names in italics (ll 570, 574, 576, 585) . Done

– l548: Foraminifera. Done

– l612: non-italic (d'Orbigny) . Done

– l620: Verlag. Done

– l643: trade-offs in. Done

– l659: Menden-Deuer. Done

– l660: . or , before journal name?  Done

– l667: Müren. Done

• Figs. 8, 9: Temperature Done

• Fig. 9: this is for the adult, not for the prolocular stage, right? please explain the different colour shading for the prey palatability. Done

• l827: sizes. Done

• l848: Eq. S2, S3 are A2, A3? Thank you for the valid point. Indeed Eq. S2, S3 are A2, A3, and this this mistake is now corrected (l906, l925)

• l852: via the Monod. Done

• l857: half-saturation. Done

• l861: reflects instead of is represented. Done

• l862: Prochlorococcus in italics. Done

• l863: for the remaining (not rest) . Done

• l876, 902: is PV Z the same as PorZ? We have corrected this mistake in the manuscript (l926)

• l889: on the Redfield; which ratio do you use? We used the 106:16 Carbon: Nitrogen Redfield ratio. This clarification now has been included (l941).

• l892: K in italics We changed the format of the parameters in the manuscript and now all the parameters are consistently not written in italics.

• l893: using Mayzaud. Done

• l899: account for the. Done

• l900: Kiørboe. Done

• l905: ) as a; size (Gentleman. Done

• l965: Within the coloured. Done

**Figure and Table layout:**

• Tables 1-2: enhance clarity with the first column left justified and text on one line if possible. Done

• Figs. 4-7: maybe use shading to identify the 'plausible' range of simulations? We changed the colour scale and added black edges for the 'plausible' and 'low biomass' symbols to make the figures easier to read.

• Figs. 8, 9: please mention the size ranges for pico, nano, micro here again

The following sentence has been included in figures' 8 and 9 legend (l856-857, l866-867): *"Six pico- (0.6-2.0 µm), ten nano- (2.6- 20 µm) and nine micro- groups (25-160 µm) are included in the model set up."*

---

## Author Response (AR2)

We would like to thank the editor and two referees for accepting our manuscript for publication. We also thank them for their useful comments which helped us to improve our manuscript. We address the editor's comments directly with our responses to the referee's reviews. Please find below our reply to the referees.

**Referee 1**

1 General comments

Thank you for considering my suggestions, and providing more detail regarding the model formulations and clarifying your assumptions. I have only two minor additions and a few typos.

2 Specific comments

1. l168: please change to: "Holling type II response with a prey refuge" or similar. Done

2. l958/959: is the important point here not that the Holling type III has grazing rates approaching zero at the lowest prey densities? Both type II and type III satiate at high prey densities

We can see that the description of how the prey refuge can be confusing. In our model, we use Holling type II as the main grazing function to describe the grazing rate of zooplankton, which satiates at high prey densities. The prey refuge term reduces the main grazing function at low density to mimic a Holling type III response. We changed the description as follow:

*"In our study, we include a prey refuge term which is based on prey's size and density based the Mayzaud and Poulet's function (1978) (Eq. (A5)). The prey refuge term describes how predators' grazing rate changes with prey density and never satiates (Gentleman and Neuheimer, 2008). At high prey density the grazing rate is similar to Holling type I, where it becomes linearly related to the prey availability ($F_{N,j_{pred}}$) (Fig. A1, Eq. (A5)). When the prey density is low, the decay constant parameter ($\Lambda$) decreases the grazing pressure such that the grazing rate is similar to Holling type III (Figure A1) (Gentleman et al., 2003). In our model, the prey refuge term causes a reduction of the grazing pressure when prey's density is low (Fig. A1)."* (l953-960).

3 Technical corrections

• l267: not very well Done

• l412: Menden-Deuer Done

• l441: pellets Done

• l775: in Tab. 2: "Phytoplankton maximum N quota" instead of "minimum"? Done

• l845: captions of Figs. 4-7 still refer to "total", adapt to "other" Done

• l835: for the remaining environments. Done

• l919: i.e. (missing dot) Done

• l928: Kjpred is named Kzoo in the equation Done

• l932: expresses Done

• l941: (106:16 molC : molN) Done

- l955: changes ... and never satiates ... Done

- l958: such that Done

- l974: shown Done

- l976: multiple prey types Done

- l995: equal to Done

- l1045: we (investigate?) the coexistence Done

**Referee 2**

This revised version fully addresses all the comments of the two reviewers in a very solid and convincing way. The authors have done a nice job for making the model description (both in the main text and in the appendix) more comprehensive. The figures 4 to 7 and the table 3 are also clearer.

I just wonder if there is an error in the equation A5 of the prey refuge term. Indeed, as indicated in the legend of Figure A1, as well as in Mayzaud and Poulet (1978) and in Gentleman and Neuheimer (2008), it seems that the multiplication term F_jprey is missing in equation A5. I also wonder why the term "phi" was not kept in this equation, as it appears in the first version of the manuscript (preference of the zooplankton for feeding on phytoplankton or zooplankton, such as phi_P + phi_Z = 1).

Except from this technical correction, I recommend to accept this article.

There is no error in the equation A5. Our prey refuge term is the same as in the Mayzaud and Poulet's function (Prey refuge $= 1 - e^{-\Lambda F}$). What is different is the way that the grazing term is defined, where we consider a more mechanistic system with temperature and food limitations. To avoid this confusion, we rephrased the Figure A1 caption to:

"Figure A1: Zooplankton grazing on one prey with and without the prey refuge term included. Prey refuge $= (1 - e^{-\Lambda F})$ (Mayzaud and Poulet, 1978). Grazing without prey refuge: $G = G_{max} * \gamma_T * \frac{F}{F + K_{jpred}}$ . Grazing with prey refuge included: $G = G_{max} * \gamma_T * \frac{F}{F + K_{zoo}} * \text{Prey refuge}$. Temperature limitation ($\gamma_T$), prey palatability ($\varphi$) and prey refuge constant ($\Lambda$) equal to 1, and $F = \varphi * B$. " (l138-144).

Regarding the "switching" term, we clarify that the revised version is the correct one and that the "switching" factor ($\Phi_{P,Z}$) is independent from the prey refuge term. In the revised version, we use the symbol "$\Phi_{P,Z}$" instead of "$\Phi_{P \text{ or } Z}$" as we believe it describes better the "switching" term of grazing equation.

[revised manuscript text omitted]